



# Upper tropospheric CO and O3 budget during the Asian Summer Monsoon

B. Barret[1,2], B. Sauvage[1,2], Y. Bennouna[1,2], and E. Le Flochmoen[1,2]

[1]Laboratoire d'Aérologie/OMP, Université de Toulouse, Toulouse, France.
[2]CNRS UMR 5560, 14 avenue E. Belin, 31400 Toulouse, France.

*Correspondence to:* Barret Brice
(brice.barret@aero.obs-mip.fr)

**Abstract.** During the Asian Summer Monsoon, the circulation in the Upper Troposphere-Lower Stratosphere (UTLS) is dominated by the Asian Monsoon Anticyclone (AMA). Pollutants convectively uplifted to the upper troposphere are trapped within this anticyclonic circulation that extends from the Pacific Ocean to the eastern Mediterranean basin. Among the uplifted pollutants are ozone

($O_3$) and its precursors, such as carbon monoxide (CO) and nitrogen oxides (NOx). Many studies based on global modelisation and satellite data have documented the source regions and transport pathways of primary pollutants (CO, HCN) into the AMA. Here, we aim to quantify the $O_3$ budget by taking into consideration anthropogenic and natural sources. We first use CO and $O_3$ data from the Metop-A/IASI sensor to document their tropospheric distributions over Asia, taking advantage of

the useful information they provide on the vertical dimension. These satellite data are used together with MOZAIC/IAGOS tropospheric profiles recorded in India to validate the distributions simulated by the global GEOS-Chem chemistry transport model. Over the Asian region, UTLS monthly CO and $O_3$ distributions from IASI and GEOS-Chem display the same large-scale features. UTLS CO columns from GEOS-Chem are in agreement with IASI, with a low bias of 11±9% and a correlation

coefficient of 0.70. For $O_3$, the model underestimates IASI UTLS columns over Asia by 14±26% but the correlation between both is high (0.94). GEOS-Chem is further used to quantify the CO and $O_3$ budget through sensitivity simulations. For CO, these simulations confirm that South-Asian anthropogenic emissions have a more important impact on enhanced concentrations within the AMA (∼25 ppbv) than East-Asian emissions (∼10 ppbv). The correlation between enhanced emissions over the

Indo-gangetic-Plain and monsoon deep convection is responsible for this larger impact. Consistently, South-Asian anthropogenic NOx emissions also play a larger role in producing $O_3$ within the AMA (∼8 ppbv) than East-Asian emissions (∼5 ppbv) but Asian lightning produced NOx are responsible for the largest $O_3$ production (10-14 ppbv). Stratosphere to Troposphere Exchanges (STE) are also important in transporting $O_3$ in the upper part of the AMA.





## 1  Introduction

Tropospheric $O_3$ plays an important role in determining the radiative budget of the atmosphere and has a non-negligible impact on climate change. In particular, according to Shindell et al. (2006), the fast economic growth of developing countries has led to an increase in tropospheric $O_3$ which may be responsible for the fast warming observed in the tropics over the latest half of the 20th century. Based on GCM simulations, Chen et al. (2007) have also shown that the changes in tropospheric $O_3$ predicted for the 21st century are likely to increase the atmospheric radiative forcing throughout the troposphere but more specifically in the tropical Upper Troposphere-Lower Stratosphere (UTLS). The understanding of the $O_3$ budget in this atmospheric region is therefore an important issue to better address future tropospheric $O_3$ radiative forcing.

During boreal summer, the northern-hemisphere extra-tropical tropospheric circulation is dominated by the Asian Summer Monsoon (ASM), which is characterized by a strong south-westerly flow in the lower troposphere converging over south and south-east Asia and results in deep convective activity over this region. During the ASM, an upper level anticyclonic circulation, the Asian Monsoon Anticyclone (AMA), builds up in response to deep convection (Hoskins and Rodwell, 1995; Garny and Randel, 2013). Based on CO UTLS data provided by the Aura/MLS (Microwave Limb Sensor) sensor, Li et al. (2005); Park et al. (2007); Barret et al. (2008) have shown that during the ASM, polluted air-masses were convectively uplifted to the UTLS and trapped within AMA circulation. Based on ACE-FTS data, Park et al. (2008) have also pointed out that similarly to CO, HCN is trapped within the AMA. Randel et al. (2010) have further highlighted that HCN from the AMA is uplifted into the stratosphere within the ascending branch of the Brewer-Dobson circulation. Data from the AIRS sensor were also used to show that the uplift of $O_3$-poor and $H_2O$ rich air masses from the Planetary Boundary Layer (PBL) is responsible for low $O_3$ within the AMA (Randel and Park, 2006). The AMA therefore appears to be an isolated atmospheric region with its physical properties and its composition likely little impacted by emissions and processes from remote regions.

Recent studies based on transport modeling have tried to determine the origin of the air masses convectively uplifted and trapped within the AMA. For instance, based on Lagrangian dispersion modeling forced with a set of reanalyzes from different systems, Bergman et al. (2013) argue that PBL air masses impacting the AMA are uplifted within a conduit centered over Northeastern India, Nepal and southern Tibet. Using high resolution WRF meteorological forcing for back trajectory simulations, Heath and Fuelberg (2014) have demonstrated that most of the air parcels convectively uplifted from the PBL and ending up in the AMA at 100 hPa originate in the Tibetan Plateau or the Himalayan southern slopes. Nevertheless, these studies based on Lagrangian modeling are not able to document the origin of pollutants in the AMA, which depends on the distribution of their sources. It is noteworthy that convection from the Tibetan Plateau, highlighted as predominant to fill





the AMA by the cited studies (Bergman et al., 2013; Heath and Fuelberg, 2014), probably plays a minor role in the transport of pollutants due to its very low pollution sources. From simulations with a global Chemistry Transport Model (CTM), Park et al. (2009) have highlighted that most of the CO

trapped within the AMA at 100 hPa comes from India and Southeast Asia and to a lesser extent from Eastern China. A more recent study based on similar simulations with the WRF-Chem limited area model comes to similar conclusions (Yan and Bian, 2015). According to Park et al. (2009), almost no CO originates from the Tibetan Plateau. Also based on CTM simulations, Li et al. (2005) point to North-East India and South-West China as the origin of upper tropospheric CO trapped within the

AMA.

Based on CTM sensitivity simulations, Kunhikrishnan et al. (2004) have quantified the impact of surface NOx from India and the neighboring regions on the $O_3$ budget over India. Their results show that $O_3$ in the Indian middle-upper troposphere (500-150 hPa) during the monsoon is mostly pro-

duced by regional (Indian) NOx emissions uplifted by convection. In particular, they point to a larger impact of NOx local surface sources relative to the lightning produced NOx (LiNOx) source on the NOx concentration in the 500-150 hPa layer during the monsoon. Based on in-situ data recorded at the Himalayan NCO-P observatory, Cristofanelli et al. (2010) have shown that high altitude $O_3$ has a marked seasonal cycle with a maximum of around 60 ppbv during the pre-monsoon season

and a minimum of 40 ppbv during the monsoon season. They show that this annual cycle is largely related to Stratosphere to Troposphere Exchanges (STE) which occur about 20% of the time all year round except during the monsoon season. During the October-May period, the Subtropical Westerly Jet (SWJ) is located between 25°N and 30°N promoting deep STE over the southern Himalayas. During the ASM, the SWJ is pushed northwards of the Tibetan Plateau by the AMA, and STE to the

central Himalayas are blocked.

Previous studies (Kunhikrishnan et al., 2004, 2006) have therefore dealt with the $O_3$ budget in the Indian troposphere, but the $O_3$ budget of the AMA has not yet been addressed in detail. In particular, it is not yet known to what extent the different NOx sources are responsible for an increase in

the $O_3$ concentrations within this upper level large scale circulation characterized by rather low $O_3$ concentrations. Furthermore, satellite data from the IASI sensor have been available since 2007 but they have not yet been used to document the ASM. These data are complementary to MLS data that have been extensively used in the region (Park et al., 2009; Barret et al., 2008) because, although they have a coarse vertical resolution, they cover both the troposphere and the UTLS. Here, we aim

to characterize the impact of STE and of NOx emissions from the different sources and regions on the $O_3$ budget in the south Asian UTLS during the monsoon season. We also use CO as a tracer of surface pollution that brings direct information about the origin of the air masses. We focus on the AMA in order to determine the role of its dynamical structure and isolation upon the regional





upper tropospheric $O_3$ budget. The second section of this paper is dedicated to the description of
the observations (IASI and IAGOS) and of the chemistry transport model GEOS-Chem (GC) that
are used in our study. In section 3, we make use of IASI and IAGOS $O_3$ and CO data to validate
their distributions simulated by the GC model over Asia. In section 4, we discuss the dynamical and
chemical characteristics of the AMA and the role of convection in controlling the distributions of
CO and $O_3$ during the ASM. Finally, the model is used to determine the impact of regional pollution
uplift, LiNOx and STE upon the CO and $O_3$ concentrations within this upper level AMA in section
5 and section 6 provides a summary and conclusions.

## 2   Observations and model

### 2.1   IASI $O_3$ and CO observations

The IASI instrument has been developed to fly on board the MetOp polar-orbiting platforms. The
first 2 platforms, MetOp-A and B, were successfully launched in 2006 and 2012 respectively. IASI
is a nadir viewing Fourier transform spectrometer observing the Earth-atmosphere Thermal Infrared
Radiation (TIR) in the 645-2760 $cm^{-1}$ wavenumber region (see e.g. Clerbaux et al. (2009)) with a
resolution of 0.5 $cm^{-1}$ after apodization. IASI provides global Earth coverage twice a day, with an
overpass time at ~9.30 and ~21.30 local time, and a pixel size on the ground of 12 km at nadir.

IASI's primary objective is the delivery of accurate meteorological products to help to improve
operational weather predictions. The IASI sensor can also monitor the tropospheric content of atmo-
spheric trace gases such as $O_3$ (Eremenko et al., 2008; Barret et al., 2011) and CO (George et al.,
2009; De Wachter et al., 2012). In the present study, we use data provided by the Software for a
Fast Retrieval of IASI Data (SOFRID) presented in Barret et al. (2011) for $O_3$ and in De Wachter
et al. (2012) for CO. In their study, Barret et al. (2011) showed that IASI enabled the independent
retrieval of $O_3$ in the lower-middle troposphere (surface-225 hPa) and in the UTLS (225-70 hPa) in
the tropics. Moreover, comparisons of SOFRID-$O_3$ data with data from $O_3$ sondes have shown that
the agreement is especially good for the UTLS column (225-70 hPa) with correlation coefficients of
0.8 (resp. 0.95) and biases of 17.5±20% (resp. 10± 10%) in Dufour et al. (2012) (resp. Barret et al.
(2011)). The ability of SOFRID-$O_3$ to capture $O_3$ daily variations in the tropical upper troposphere
has also been demonstrated and validated against IAGOS cruise data in Tocquer et al. (2015). The
SOFRID CO data have been validated against MOZAIC data in De Wachter et al. (2012). SOFRID
data are able to capture the seasonal variability of CO at mid-latitudes (Frankfurt) as well as at trop-
ical latitudes (Windhoek) in the lower (resp. upper) troposphere with correlation coefficients of 0.85
(resp. 0.70). At Windhoek, in the lower (resp. upper) troposphere SOFRID-CO data are biased low





with $13\pm20\%$ (resp. $4\pm12\%$) compared to MOZAIC data.

## 2.2 IAGOS $O_3$ and CO observations

The MOZAIC program was set up to provide routine measurements of reactive gases on long distance commercial aircraft (Marenco et al., 1998). In 1994, five airliners were equipped with $O_3$ and relative humidity instruments, and a CO analyzer was successfully added in December 2001. Since 2009, the MOZAIC program has been expanded, implementing other commercial in-service aircraft observation programs such as CARIBIC (http://www.caribic-atmospheric.com) into the IAGOS (In-service Aircraft for a Global Observing System) Research Infrastructure. The MOZAIC/IAGOS data are freely accessible for scientific use at http://www.iagos.org. MOZAIC/IAGOS measurements carried out with a 30 s response time and with a reported precision of $\sim 5$ ppbv for CO (Nedelec et al., 2003) and 1 ppbv for $O_3$ (Thouret et al., 1998) have an horizontal resolution of about 7 km at cruise altitude and a vertical resolution of about 300 m during ascents and descents (Nedelec et al., 2003). For the present study, we used IAGOS profiles measured at take-off and landing near Hyderabad ($17.2°N$, $78.3°E$) in central India from May to October 2009. CO data were available for each month but for $O_3$, no data were produced in September and October following an instrument failure. For both gases, we could use from 10 to 16 profiles for each month with available data.

## 2.3 GEOS-Chem configuration

In order to compute the CO and $O_3$ budgets in the Asian upper troposphere, we use the GEOS-Chem (GC) global chemistry transport model (Bey et al., 2001) version 9-01-01 with a set-up similar to that described in Yamasoe et al. (2015). This model has been thoroughly evaluated over the tropics through comparisons with in situ and remote sensed measurements of $O_3$, CO, $NO_2$ and $HNO_3$ (e.g. Martin et al. (2002) ; Sauvage et al. (2007a) ; Yamasoe et al. (2015) ). GC is driven off-line by the meteorological analyses from the Goddard Earth Observing System (GEOS-5) of the NASA Global Modeling and Assimilation Office (GMAO). Tropospheric chemistry includes both $O_3$-NOx-hydrocarbons and aerosols chemistry. Stratospheric $O_3$ chemistry is computed with the linearized Linoz stratospheric ozone scheme developed by McLinden et al. (2000). STE are diagnosed with tagged $O_3$ simulations including a stratospheric $O_3$ tracer. Convection is parameterized with the relaxed Arakawa-Schubert scheme (Moorthi and Suarez, 1992) in GEOS-5. Turbulent mixing in the planetary boundary layer is described in Wu et al. (2007). The simulations are performed on a regular $2°\text{x}2.5°$ horizontal grid and on 47 hybrid pressure-$\sigma$ levels from the surface up to 0.01 hPa. Emissions from Biomass Burning (BB) come from the monthly Global Fire Emissions Database version 2 (GFED-v2) (van der Werf et al., 2010). The global anthropogenic emissions are taken from the EDGAR v.4.1 inventory which provides annual global emissions of greenhouse gases and





ozone precursors on a 1° x1° horizontal grid, but typically overwritten by data from various regional inventories. For instance over Asia we use the detailed inventory from Streets et al. (2006). Re-
gional emission inventories are also used over Europe (EMEP), Canada (CAC), Mexico (BRAVO) and North America (EPA/NEI99 with ICARTT modification). All anthropogenic inventories are scaled for the year 2005. Biogenic emissions are taken from MEGAN v2.1. Detailed information on these emission inventories can be found on http://acmg. seas.harvard.edu/geos/doc/archive/man.v9-01-01/index.html. NOx emissions from lightning are computed according to cloud top height pa-
rameterization (Price and Rind, 1994), rescaled with LIS-OTD climatology (Sauvage et al. (2007a) ; Murray et al. (2012)) and are estimated at almost 6 Tg(N)/year (Martin et al., 2007).

We have performed 11 simulations for the May to October (MJJASO) period of 2009 with a six month spin-up. The control run was performed with all the emission sources considered. In order to determine the relative importance of the different sources on the CO and $O_3$ Asian UTLS budgets,
we have performed sensitivity runs with emissions alternatively switched off. For CO, the sensitivity simulations concern South (0-40°N,60-100°E), East (15-40°N, 100-125°E) and South-East °S-15°N, 100-150°E) Asian anthropogenic and African (20°S-20°N, 20°W-50°E) BB emissions. For $O_3$, we considered the impact of NOx surface emissions from the same sources as for CO, and of LiNOx emissions from the two monsoon regions, South-Asia (0-40°N, 60-100°E) and Africa
(20°S-20°N, 20°W-50°E). In order to investigate the stratospheric contribution on the AMA tropospheric ozone budget (section 5.2), we use a tagged ozone tracer to follow the stratospheric ozone flux across the tropopause as used in Sauvage et al. (2007b) and described by Fiore et al. (2002). The tagged simulation submits ozone produced in different regions of the atmosphere to archived three-dimensional fields of production and loss frequencies, allowing tropospheric ozone to be de-
constructed into components from stratosphere and troposphere. The results from the sensitivity simulations are described and analyzed in section 5.1 for the CO budget and in section 5.2 for the $O_3$ budget.

### 2.4    IASI and GEOS-Chem comparisons

In order to validate the CO and $O_3$ distributions simulated by the GC model, we use SOFRID CO and $O_3$ retrievals to have a regional view of these distributions. The comparisons are made for monthly averaged profiles on the 2°×2.5° GC grid. The GC profiles are first interpolated on the 43 vertical retrieval levels from SOFRID. IASI vertical profiles have a vertical resolution (∼6-8 km) that is much lower than those modeled by GC (100 m to 1 km). In order to take these resolution differences
into account and make a sound comparison, we have to convolve the GC vertical interpolated profiles



with IASI averaging kernels (AvK) according to the classical smoothing equation (e.g. Barret et al.
(2005); De Wachter et al. (2012); Liu et al. (2009)):

$$\hat{\boldsymbol{x}}_{GC} = \boldsymbol{x}_a + \mathbf{A} \cdot (\boldsymbol{x}_{GC} - \boldsymbol{x}_a) \tag{1}$$

where $\boldsymbol{x}_{GC}$ and $\hat{\boldsymbol{x}}_{GC}$ are the original and the smoothed or convolved GC profiles. $\mathbf{A}$ is the SOFRID
AvK matrix which describes the sensitivity of the retrieved to the true profile (see Rodgers (2000) for
a description of the AvK matrix) and $\boldsymbol{x}_a$ is the a priori profile used for the retrieval (the description
of the a priori profiles can be found in Barret et al. (2011) for $O_3$ and De Wachter et al. (2012) for
CO).


## 3    Modeled versus observed CO and $O_3$ distributions

The comparisons of the tropospheric $CO/O_3$ Asian distributions simulated by GC and observed by
IASI enable us to evaluate the model's capacity to reproduce the large scale features of the distribu-
tions and the possible causes of discrepancies. Airborne IAGOS profiles measured in central India
will provide a more precise evaluation of the absolute values simulated locally by the model.

### 3.1    CO in the Asian troposphere

The monthly distributions of UTLS (270-110 hPa) CO columns from IASI and GC are displayed in
Fig. 1 for the region extending from Africa to Indonesia and from 10S$°$ to 40$°$N for the May (pre-
monsoon) to October (post-monsoon) period. The dominant features of these distributions are the
maxima over Africa and Asia. The statistics of the CO UTLS columns comparison (for the domain
and the 6 months displayed in Fig. 1) are summarized in Table 1. GC understimates the columns by
11±9% relative to IASI with a correlation coefficient of 0.70. The smoothing has little impact on
the bias but reduces the relative standard deviations of the differences and enhances the correlations.
The comparison between GC simulations forced with GEOS-5 analyses and MLS at 215 hPa for the
tropical band of Liu et al. (2013) gives similar results with a 10 ppbv bias and a correlation coeffi-
cient of 0.65.

Over Africa the observed maximum shifts from western Africa in May to Central and Southern
Africa in July and September following the BB season (Sauvage et al., 2005) . We notice that the GC
upper tropospheric CO distributions over Africa display the same kind of discrepancies with IASI
than those shown by Liu et al. (2010, 2013) with MLS. Indeed, their GC simulations have CO con-
centrations that are systematically too low at 215 hPa over Central Africa in July (Liu et al., 2013)





and from August to October (Liu et al., 2010). Furthermore, Barret et al. (2010) have shown that
5 CTMs using GFEDv2 for BB emissions underestimate the upper tropospheric CO concentrations
during the monsoon over Africa between 10°S and 5°N by up to 50 ppbv compared to MOZAIC
in-situ data. The use of the BB emission inventory from Liousse et al. (2010) leads to a correction of
these biases and even to an overestimation of modeled upper tropospheric CO over Africa. The bias
documented here probably results from too low BB emissions over central and southern Africa from
GFEDv2. Nevertheless, African BB emissions are not expected to impact the AMA composition and
the observed biases will not impact our results.

Over Asia, which is the focus of our study, the highest CO columns are simulated by GC and
detected by IASI over East Asia before the monsoon (May), over the continental convective region
corresponding to Northern India, Nepal and southern Tibet during the monsoon (JJA) and back over
East Asia after the monsoon (September-October). We have used a threshold of 2.5 kg/m$^2$/s for the
upward convective mass flux from the GEOS-5 analyses in the upper troposphere (350-150 hPa) to
identify the deep convective areas (see contours in Figure 1). The ASM region is indeed characterized
by GEOS-5 upward convective mass flux values comprised between 1 and 5 kg/m$^2$/s in the upper
troposphere (not shown) and 2.5 kg/m$^2$/s corresponds to relatively strong convective uplift. During
July and August, high CO UTLS columns are also captured by the model and IASI within the AMA,
as has already been documented in Park et al. (2009) and Barret et al. (2008). The AMA is delimited
by the 12520 m GH contour at the 200 hPa level, as done in Randel and Park (2006) (see section 4.1
for the definition of the AMA boundaries). More specifically, IASI detects enhanced CO columns in
agreement with raw GC columns over the monsoon region and underestimates the CO columns in
the western part of the AMA. This is an effect of IASI's limited vertical sensitivity, as appears from
the GC UTLS distributions once the profiles are smoothed by IASI averaging kernels according to
equation 2.4 (Fig. 1 second column) resulting in lower UTLS columns and a better agreement with
IASI. This is confirmed by the longitude-pressure cross-sections averaged over the 21-29°N band
that correspond to the southern part of the AMA (Fig. 2), where we notice that the AvK smoothing
mixes the UTLS enhanced concentrations throughout the middle and upper troposphere, leading to
a better agreement with IASI cross-sections. In the eastern part of the AMA, CO UTLS concen-
trations are higher and better detected by IASI, resulting in a lesser effect of the smoothing and a
better agreement between IASI and GC raw columns. Our results apparently disagree with Liu et al.
(2013), who report larger underestimations of UTLS GC CO over Asia than elsewhere in July 2005
especially at 100 hPa. They argue that this model underestimation probably results from insufficient
convective uplift to 100 hPa with GEOS-5. Indeed, our comparisons with IASI (Fig. 1) do not show
enhanced underestimation of the GC UTLS columns in the Asian region and in the AMA. The low
vertical resolution of IASI and its lack of sensitivity above 150 hPa highlighted in Fig. 2 are probably





responsible for this apparent contradiction with Liu et al. (2013).

   The good agreement of IASI and GC in the middle and upper troposphere within the enhanced CO region is confirmed by looking at the latitude-pressure cross-sections averaged over the 75-105°E longitude domain where convection is active (Fig. 3). IASI clearly detects the UTLS enhanced CO

concentrations between 400 and 200 hPa resulting from convective detrainment in very good agreement with GC. Kar et al. (2004) have already shown that the MOPITT sensor was able to detect UTLS CO enhancements disconnected from the lower troposphere and resulting from convective detrainment during the ASM. Our IASI latitude-pressure cross sections clearly show that IASI is also able to detect such CO UTLS bubbles. Both IASI and GC document that the southern edge

of the CO enhancements shifts from 10 to 20°N from May to July and back to 10°N from August to October (Fig. 3). Nevertheless, GC underestimates CO throughout the troposphere around 15°N particularly in May-June. These results are confirmed by CO profiles measured by the IAGOS programme in Hyderabad (Fig. 4). In the middle and upper troposphere, the agreement between IAGOS and GC is within the $1\sigma$ variability, except during the May-June period, characterized by an impor-

tant CO underestimation by GC, with however good modeling of the CO seasonal variation. Finally, enhanced UTLS CO columns from August to October over Indonesia also correspond with CO enhanced concentrations between 500 and 200 hPa at the Equator in both IASI and GC distributions in Fig. 3.

Even if the focus of our study is the upper troposphere, we note that during the May-October period, high CO concentrations are detected by IASI and simulated by GC in the lower and middle troposphere within the monsoon polluted region over 20-35°N (Fig. 3) and 70-120°E (Fig. 2). Enhanced CO concentrations (> 110 ppbv) are also detected by IASI west of 70°E over the Middle East and northern Africa (Fig. 2) where the model simulates lower CO concentrations even if

the model-satellite bias is partly corrected when smoothing by the AvK is taken into account. The smoothing is responsible for mixing high CO concentrations simulated close to the surface to the lower and free troposphere. The discrepancy between GC and IASI in the free troposphere is larger between May and August than in September-October. The study of Liu et al. (2010) also documents an underestimation by GC of TES for CO at 681 hPa over the Middle-East and northern Africa

that is larger in August than in September and October 2005 (see their Figure 3). The underestimation of CO by the GC in the lower and middle troposphere also appears south of 20°N in Fig. 3. Comparisons between GC and IAGOS profiles in Hyderabad (Fig. 4) confirm these overly low CO concentrations simulated by GC below 600 hPa with decreasing differences from June to October.

Concerning the upper troposphere, both GC and IASI are able to capture the seasonal variability associated with the ASM and particularly the CO enhancements within the AMA. It is noteworthy





that IASI enables the detection of uplifted CO in the ASM region. Nevertheless, GC significantly underestimates CO in the lower and middle troposphere during boreal spring over India compared with IASI and IAGOS.


## 3.2 O$_3$ in the Asian troposphere

Concerning O$_3$ GC versus IASI comparisons, it is important to note that using equation 2.4 to smooth GC profiles implies mixing stratospheric O$_3$ concentrations in the UTLS column. The averaging kernels displayed in Barret et al. (2011) show for instance that the O$_3$ concentration retrieved at 150

hPa is sensitive to O$_3$ up to about 50 hPa. Stratospheric biases in the model would therefore imply an apparent bias in the modeled UTLS column compared with IASI. As mentioned above, we use GC version 9-01-01 which stratospheric O$_3$ is based on the linearized scheme from McLinden et al. (2000). Recently Eastham et al. (2014) have evaluated stratospheric O$_3$ from GC version 9 (using Linoz) versus a new version (not publicly available at the time of this study) using the Uni-

versal tropospheric-stratospheric Chemistry eXtension (UCX). They show that, averaged annually, GC-Linoz total columns of ozone are biased by 25 to 50 DU compared with TOMS in the band from 40°S to 40°N. The annual averaging hides much larger regional and seasonal discrepancies. Indeed, from their Fig. 2 we can roughly estimate that for the May-October period of interest here, the overestimation of the total columns can reach 100 DU in the tropics and in the southern hemi-

sphere, down to 60°S. From Dufour et al. (2012), we also know that SOFRID stratospheric O$_3$ is highly biased compared to ozonesondes with biases of 8±5% for the column up to 30 km and 7±5% for the stratospheric (16-30 km) column. Comparisons between IASI and GC for the May-October period in the 30°S-30°N band show that the mean GC stratospheric (90-24 hPa) column is 1.66 times higher than IASI mean column. Taking the 7% IASI bias in the tropics into account, we have

applied a 0.58 scaling factor to GC profiles in the lower and middle stratosphere (90-24 hPa) before applying the AvK smoothing.

The UTLS O$_3$ columns are displayed in Fig. 5. The most obvious feature of the distributions captured by IASI and GC is the transition from low columns in the tropical UT south of the tropopause

(2PVU) to high columns in the extra-tropical lower stratosphere. This transition closely follows the undulation of the tropopause. From June to September, the tropopause is pushed northwards by the AMA circulation and the region from the Middle East to East Asia is characterized by intermediate O$_3$ columns. The region of lowest O$_3$ columns is simulated and observed over the western Pacific in May and progresses northwestwards to South-East Asia and South India until October. Over Africa,

IASI and GC document a southward shift of moderate O$_3$ columns from western Africa in May to southern Africa in September-October. This general good agreement between IASI and GC O$_3$ distributions translates into correlation coefficients higher than 0.9 and a mean bias of 14±26% (see





Table 1). Biases between IASI-SOFRID and UTLS columns from ozonesondes were estimated to be 17.5±19.3% (Dufour et al., 2012) and 10±10% (Barret et al., 2011) once the ozonesondes profiles

were smoothed by IASI AvK. The mean value of the GC UTLS columns over our study region is therefore most likely to be in good agreement with ozonesondes. The good behavior of GC UTLS $O_3$ is corroborated by comparisons between GC and IAGOS profiles at Hyderabad which show a very good agreement between the surface and 200 hPa during the May-August period (see Fig. 4). Unfortunately, no $O_3$ data are available from IAGOS Hyderabad-Frankfurt flights in September and

October 2009.

When smoothing is applied to GC profiles, the features of the $O_3$ distribution remain similar but some corrections are introduced. Over most of the domain, the GC UTLS columns are slightly increased leading to a better agreement with IASI with differences within ± 50%. On the other hand,

over the oceanic convective regions of the western Pacific characterized by the lowest $O_3$ absolute values, the smoothing tends to decrease the UTLS column leading to the highest relative biases (exceeding -50%). This decrease of UTLS $O_3$ when IASI AvK are applied has already been reported in Dufour et al. (2012) for ozonesonde profiles as a result of the accentuation of the $O_3$ S-shape for tropical profiles. The effect is therefore more important for convective oceanic profiles which have

the most marked S-Shape.

The latitude-pressure cross sections displayed in Fig. 6 highlights the impact of the convolution of the modeled profiles by IASI AvK to smooth the lower stratosphere to upper troposphere transition and to decrease the height of the chemical tropopause. The very low $O_3$ concentrations from the

model smoothed profiles over the Bay of Bengal convective region (south of 20°S) result from the accentuation of the S-shape profiles discussed above. These cross-sections also indicate the northwards shift of the tropopause and of high UTLS (300-150 hPa) $O_3$ concentrations from May until September. It is interesting to note the large $O_3$ concentrations originating from the stratosphere in the middle troposphere down to 700 hPa between 20 and 30°N in May and June that almost dis-

appear in July and August, only to reappear in October. The seasonal variations of STE that both model and observations are pointing to are in good agreement with the results from Cristofanelli et al. (2010) which, based on in-situ data in the Himalaya, indicate the absence of stratospheric intrusions during the monsoon season.

Fig. 7 presents the $O_3$ longitude-pressure transects over Asia. In the middle-upper troposphere, both model and observations display a persistent west-east gradient with lower $O_3$ concentrations east of 70°E. This gradient is the highest during the Asian monsoon period when convection is the most active in the western part of the domain and when the Middle-East is characterized by its annual $O_3$ maximum (Li et al. (2001); Liu et al. (2009, 2013)). Nevertheless, from June to September,



the UT $O_3$ concentrations are not homogeneously low in the convective region and enhanced $O_3$ concentrations are simulated and observed between 100 and 120°E. In the model, the lowest UT $O_3$ concentrations coincide with the deepest convection centered around 75°E, and the enhanced concentrations coincide with less intense convection, as illustrated by the 2.5 kg/m$^2$/s convective upward mass-flux contour.

The general features of the tropospheric and UTLS $O_3$ distribution over the large Asian region simulated by the GC are in good agreement with those observed by IASI. The application of the AvK convolution to GC vertical profiles decreases the altitude of the chemical tropopause, smoothes some of the modeled high resolution features and accentuates the S-shape of convective oceanic $O_3$ profiles. Nevertheless, the model and IASI display the same longitudinal and latitudinal gradients,

both in the middle and in the upper troposphere over Asia.

## 4    Dynamical and chemical characterization of the AMA

The first part of this section is dedicated to the characterization of the AMA as a 3D volume based on dynamical parameters to enable the quantification of chemical budgets within this upper level

anticyclone (section 5). We will then discuss the dominant role played by convection in controlling tropospheric CO and $O_3$ distributions over Asia and more particularly within the AMA.

### 4.1    The Asian Monsoon Anticyclone: a 3D volume

During May and October, the convective activity mostly takes place over southeast Asia and the 150 hPa tropopause is located between 30 and 35°N over Asia and the AMA is not present (see Fig. 1). In

June and September, at the beginning and at the end of the ASM, the convective activity has moved northwards towards the Bay of Bengal and the AMA is present over northeastern South-Asia. During the heart of the ASM (July-August), the region impacted by convection encompasses the Bay of Bengal, India, Bangladesh, Nepal and southeastern Tibet and the tropopause is pushed to 40° north by the AMA which is fully developed and extends roughly from 20 to 40°N and from 30 to 120°E

and vertically from 300 to 100 hPa. The center of the AMA is bimodal with the high pressure center located alternatively over the Tibetan plateau and over Iran (Zhang et al., 2002). This high level anticyclone is characterized by large scale periodic elongations and sheddings as described in Popovic and Plumb (2001). The AMA air masses are characterized by low potential vorticity (PV) values or high geopotential heights (GH). Barret et al. (2008) and Garny and Randel (2013) have shown that

enhanced CO concentrations are strongly correlated with low PV values and are therefore controlled by the oscillations and sheddings of the AMA.





In previous publications, the edge of the AMA has been defined as constant GH contours at different pressure levels. Randel and Park (2006) (resp. Heath and Fuelberg (2014)) use a 14320 (resp. 14430) m GH for the AMA at 150 hPa and Bergman et al. (2013) use 12520 (resp. 16770) m GH at 200 (resp. 100) hPa. In order to characterize the AMA as a closed volume, we have looked for a criterion independent of the pressure level. The anticyclonic air masses are characterized by higher GH than the surrounding air masses. We have therefore tested thresholds of GH anomalies. We use the GH fields from the MERRA re-analyses which are provided on 42 levels from the surface to 0.1 hPa with a $1.25° \times 1.25°$ horizontal resolution. The anomalies are computed as the differences between the mean zonal GH computed over the $50°N$ to $50°S$ latitudinal band and the local GH. The AMA appears very clearly at different UTLS levels as the region with the highest GH anomalies on Fig. 8. The contours corresponding to a 270 m GH anomaly best match the 16770, 14320 and 12520 m GH isocontours at 100, 150 and 200 hPa corresponding to the AMA edge in Bergman et al. (2013), Randel and Park (2006) and Bergman et al. (2013) respectively. We have therefore chosen a 270 m GH anomaly as the threshold for the AMA boundary throughout the UTLS. In section 5, within the AMA and outside of the AMA both refer to the tropospheric part of these atmospheric regions bounded by the 2PVU contour.

### 4.2   Relationship between convection and the CO and $O_3$ distributions

The studies presented in the introduction have highlighted the AMA as a region with a composition that is very different from its surroundings, according to UTLS satellite observations. The use of IASI data brings information about CO and $O_3$ over the whole troposphere and therefore allows to better document the link between the upper tropospheric distributions and transport processes such as convection. In the following paragraph, we analyze the modeled and observed $O_3$ and CO distributions in light of their relationship with convection.

In the middle troposphere, the longitude-pressure sections of CO and $O_3$ presented above are anti-correlated. West of about $80°E$, in the monsoon region characterized by the strongest convective upward mass fluxes from GEOS-5, low CO (90 ppbv) is associated with high $O_3$ (60 ppbv) and East of $80°E$ high CO is associated with low $O_3$. This anti-correlation is clear both from the model outputs and from IASI data. The high summer tropospheric $O_3$ extending from western India to north Africa has been first described as the "Middle East tropospheric Ozone maximum" by Li et al. (2001) and further analyzed by Liu et al. (2009, 2010). The subsidence associated with the AMA is taking place in the middle troposphere on its western side over the Eastern Mediterranean, the Middle East and Central Asia (Hoskins and Rodwell, 1995; Liu et al., 2009). This phenomenon is clearly seen in Fig. 10 that displays GC $O_3$ fluxes in a longitude pressure cross-section at the center of the AMA. This descent of air masses impacted by Asian pollution trapped within the AMA contributes to the summer "Middle East tropospheric Ozone maximum". In their analysis, Liu et al. (2009) have



shown that the $O_3$ buildup is favored by the Arabian and Saharan anticyclones that isolate the middle

troposphere over this region. From simulations with tagged $O_3$ Liu et al. (2009, 2010) attribute an

equivalent and dominant impact (30-35%) on the $O_3$ maximum over the middle East to local sources

and transport from Asia via the UT and the AMA circulation. Over Northern Africa, transport from

Asia contributes less thann regional sources. It is clear from the CO GC distributions displayed by

Liu et al. (2009) (their Fig. 6) and from the present study as well as from our IASI data (Fig. 2 and

7) that the $O_3$ Middle East maximum in the middle troposphere coincides with relatively low CO

concentrations.

Between 80 and 120°E, the low $O_3$ and high CO concentrations result from the convective activ-

ity occurring in South and South-East Asia during the monsoon. Convection mixes CO between the

Asian polluted PBL and the upper troposphere resulting in enhanced concentrations over the whole

troposphere. The overlap between important CO sources and convection occurs primarily over the

Indo-Gangetic Plain (IGP) according to Fig. 9 that displays anthropogenic CO emissions from the

Streets inventory (Streets et al., 2006) and GEOS-5 upward convective mass fluxes. The impact

of convective transport on the $O_3$ distribution is more complicated. It results from two antagonist

effects: the vertical mixing of $O_3$ itself and the uplift of $O_3$ precursors followed by enhanced pho-

tochemical $O_3$ production (Doherty et al., 2005; Lawrence et al., 2003). The vertical mixing results

in the transport of $O_3$ poor air masses from the lower troposphere where $O_3$ lifetime is short to the

upper troposphere where it is long and $O_3$ rich air masses from the upper to the lower troposphere

by compensatory subsidence. The effect of this overturning is a decrease of UT $O_3$ and of the tropo-

spheric $O_3$ burden and lifetime. Over polluted regions, such as Asia, convection uplifts $O_3$ precursors

(especially NOx) result in an increase of the $O_3$ production in the middle and upper troposphere at

the expense of the lower troposphere. The electric activity from convective storms is responsible for

the in-situ production of LiNOx, also responsible for an increased $O_3$ production. This source of

$O_3$ clearly appears in Fig. 10 where the net $O_3$ production rates are enhanced between 500 and 150

hPa in the monsoon region. Convective clouds also diminish the tropospheric photochemical activ-

ity through a reduction of the solar UV radiations. These combined effects are responsible for the

lower $O_3$ concentrations over South Asia compared to regions with high insolation and downward

transport of $O_3$, such as the Middle East and North Africa.

In the Asian upper troposphere in June, the AMA is building up and only extending between 60

and 120°E and the $O_3$ fluxes switch from downward to upward around 90°E (Fig. 10). In July and

August, the AMA is well established over the 15-145°E domain and the upward flux remains east

of 90°E in the monsoon region while the strongest downward fluxes move to the western edge of

the AMA between 15 and 45°E. As already discussed, this downward flux largely contributes to

the build-up of the Middle East $O_3$ maximum as described in Liu et al. (2009). In September, the



situation is similar to June, the AMA has largely shrunk and the $O_3$ production is associated with
an $O_3$ downward flux between 75 and 90°E. Above the continents, the photochemistry illustrated by
the $O_3$ net production rates in Fig. 10 switches from a net source of $O_3$ in the polluted PBL, to a net
sink in the free troposphere below about 500 hPa and again to a net production in the middle and
upper troposphere. This behavior agrees with the different NOx photochemical regimes discussed
in Jacob et al. (1996). In particular, low NOx concentrations are responsible for the destruction of
$O_3$ in the lower and middle troposphere and slightly higher concentrations produce $O_3$ in the up-
per troposphere, as explained in (Brune, 1992, IGAC Report). During the whole period, the $O_3$ net
production pattern in the middle and upper troposphere is characterized by a double maximum with
values exceeding 5 ppbv/day that are associated with the upward fluxes east of 90°E and the down-
ward fluxes west of 90°E. Both upper troposphere maxima are located within the eastern half of the
AMA. Below the tropopause, the $O_3$ net production rate is exceeding 2 ppbv/day within the whole
AMA. The enhanced net $O_3$ production rates are associated with enhanced NOx concentrations (100
pptv contour in white). In the upper troposphere, the AMA therefore appears as a region of high $O_3$
production, resulting from the trapping of NOx from various sources. In the next section, we deter-
mine the impact of the different sources on the CO and $O_3$ budgets within the AMA.

## 5 CO and $O_3$ budget

Our aim here is to characterize the origin of CO and $O_3$ within the Asian upper troposphere during
the monsoon season by comparing the impact of the different emission sources inside and outside
of the AMA based on sensitivity simulations for the different type of emissions and for the different
regions of interest. For CO we have considered anthropogenic and BB emissions and for $O_3$ we have
considered the production of NOx originating from anthropogenic, BB and lightning sources and the
transport of stratospheric $O_3$ through STE.

### 5.1 The CO budget

As mentioned in section 2.3, we have considered the two main regions of importance concerning
anthropogenic CO emissions: South and East Asia. Park et al. (2009); Yan and Bian (2015) have
indeed highlighted the predominant role of Asian sources from these two regions in filling the AMA
with CO. We can also notice that the surface fluxes of CO used for our GC simulations (Fig. 9) are
the largest for the whole Asian region over north-eastern China and for the South-Asian domain over
the IGP. These fluxes are consistent with those used in Park et al. (2009) and Yan and Bian (2015).
Concerning BB, Nassar et al. (2009) have shown that Indonesian BB emissions had a large impact
on the Indian upper troposphere composition in 2006 following the perturbation of the tropical cir-
culation by a strong El Nino event. Our sensitivity simulations performed for Indonesian or South



East Asian anthropogenic sources have shown that in 2009 this region was not impacting the south
Asian upper troposphere (not shown). The simulation with African BB CO emissions switched off
also results in negligible modifications of the CO distribution in the south-Asian upper troposphere
(not shown).

The differences between the reference simulation and the sensitivity simulations with anthropogenic CO emissions from South and East Asia shut down are displayed in Fig. 11 for the pressure-longitude section (21-29°N) and in Fig. 12 for the upper tropospheric (200 hPa) distribution. The average CO mixing ratio differences between on and off simulations within and outside of the AMA are given in Table 2. The pressure-longitude sections clearly show that the upper troposphere and especially the AMA ones are more impacted by South Asian than East Asian emissions. For the 4 months considered, CO from South Asia is responsible for CO enhancements of 20 to 30 ppbv within the AMA between 300 and 100 hPa, while East Asian emissions mostly impact regions below 200 hPa on the eastern side of the AMA. This result is expected from the correlation between high emissions and strong convection over South Asia as can be seen in Fig. 9. High convective mass fluxes ($>$ 2.5 kg/m$^2$/s) at 225 hPa are located over the IGP, where CO emission fluxes exceed 150 kg/km$^2$/day. East of the Himalaya, the emissions are largest over eastern China where convection is not as strong as over the IGP. The region with the strongest South-Asian CO uplift in the middle troposphere lies between 75 and 105°E according to the GC (see Fig. 11) which is consistent with Bergman et al. (2013), who highlights that PBL air masses that reach the UTLS pass through a mid-tropospheric conduit located roughly over the same region.

In the upper troposphere at 200 hPa, East Asian emissions are only responsible for CO enhancements of about 10-20 ppbv located over south-east Asia and China during the monsoon. Larger CO enhancements are caused by South Asian emissions with the highest values ($>$ 35 ppbv) located within the convective region around 75°E and 27°N and values exceeding 20 ppbv that spread within the AMA bounded by the tropopause to the north. These values are higher than those of Yan and Bian (2015), who found CO enhancements of 12-30 ppbv from Indian sources and of 5-9 ppbv from Chinese sources at 215 hPa. At 100 hPa (not shown), East Asian sources contribute to less than 6 ppbv to UTLS CO which is slightly lower than what Yan and Bian (2015) and Park et al. (2009) have documented. Concerning South Asian sources, they are responsible for 12 to 20 ppbv CO enhancements (not shown) in good agreement with Yan and Bian (2015) and Park et al. (2009).

The average figures of Table 2 summarize these results. South Asian CO emissions are responsible for a strong CO enhancement within the AMA from June to September with a maximum of $\sim$25 ppbv during the monsoon peak in July-August. Furthermore, average CO enhancements from South Asian emissions are about 10 ppbv larger within than outside of the AMA which further



highlights the AMA as a trap for uplifted South-Asian pollution during the monsoon. East-Asian emissions result in maximum enhancements of about 10 ppbv in the UTLS during July-August. The little differences between the enhancements computed within and outside of the AMA also show

that East-Asian sources are located outside of the conduit connecting boundary layer air masses and the AMA described in Bergman et al. (2013).

## 5.2 The $O_3$ budget

The contribution to the $O_3$ burden from the main sources of NOx emissions is computed from sen-

sitivity simulations with the GC model. Sauvage et al. (2007b) have shown that tropospheric $O_3$ over Asia during the monsoon is mostly impacted by Asian sources. Focusing on the Indian region, Kunhikrishnan et al. (2006) have also highlighted the predominance of Asian sources (India, China and Indonesia) on the Indian tropospheric $O_3$ budget during the monsoon. They have also shown that Middle-East emissions have a small impact on NOx and $O_3$ concentrations below 500 hPa and

that African and Middle-East sources have a negligible impact in the middle and upper troposphere over India during the ASM. We have therefore chosen to focus on the impact of Asian emissions upon the AMA $O_3$ burden. Concerning anthropogenic emissions, we have separated Asia into the same three main regions as for CO (see section 2.3). One of the main conclusion of Sauvage et al. (2007b) is that LiNOx is the most important NOx source controlling the tropical tropospheric $O_3$

burden. We therefore performed simulations to characterize the importance of LiNOx from the two nearby monsoon regions (see 2.3) upon upper tropospheric $O_3$ during the ASM. Finally, the impact of STE was established using the GC stratospheric $O_3$ tagged tracer as explained in section 2.3.

For $O_3$, the results of the sensitivity simulations are displayed in Fig. 13 for longitude-pressure

sections averaged over the 21-29°N band and in Fig. 14 for maps at 200 hPa. The results are summarized in Table 3 for $O_3$ and NOx average mixing ratios. The enhancements of $O_3$ by NOx anthropogenic emissions from South and East Asia are closely linked to those of CO previously analyzed. As for CO, sensitivity simulations with Indonesian anthropogenic and African BB NOx sources switched off (not shown) show very little impact on South Asian upper tropospheric $O_3$.


The $O_3$ enhancements caused by East-Asian emissions is the largest (> 15 ppbv) below 300 hPa between 90 and 120°E. Convection is not strong enough over China to bring PBL NOx deep into the AMA and on average, upper tropospheric $O_3$ enhancements from Chinese emissions are about 5 ppbv both within and outside of the AMA during July-August. Compared to Chinese emissions,

South-Asian emissions have a smaller impact on free tropospheric $O_3$ (<12 ppbv) but a larger scale impact on $O_3$ in the upper troposphere and more specifically within the AMA. On average, South-Asian emissions are responsible for an $O_3$ (resp. NOx) increase of 8 (resp. 0.04) ppbv within the





AMA and of about 5 (resp. 0.015) ppbv outside of the upper-level anticyclone (Table 3). Indian NOx are uplifted and trapped within the AMA (see white contours in Fig. 14) and produce $O_3$ molecules

that are also trapped within the AMA.

Asian LiNOx are responsible for an important $O_3$ production in the Asian upper troposphere mostly confined within the AMA (see Fig. 13 (h) and (m)) with a strong intra-seasonal variability. In July, LiNOx produce 13.5 ppbv $O_3$ in the AMA and only 10 ppbv in August. In both cases, the $O_3$

production outside of the AMA is half of its value within the AMA. For NOx, the production within the AMA is about 2.5 higher than outside of the AMA, highlighting the non-linearity of the $O_3$ production by NOx. The impact of African LiNOx over Asia varies strongly from June to September. In June and September, when the AMA is weakened and located east of 90°E, African LiNOx have a large impact in the upper troposphere over the Middle-East and in the free troposphere further east

over India (Fig. 14). In July and August, the AMA circulation that extends to 30°W prevents air masses impacted by African LiNOx from affecting $O_3$ in the Middle- East upper troposphere and the free troposphere over India is also much less impacted than in June and September. During the July-August period, the large subsidence of air masses over the Middle-East (30-60°E) brings $O_3$ produced by both South Asian anthropogenic NOx and Asian LiNOx down to 500 hPa (Fig. 14) and

contributes to the mid-tropospheric $O_3$ maximum demonstrated by GC and IASI (see Fig. 7).

The last source of $O_3$ in the Asian upper troposphere that we investigated is STE. At 200 hPa, STE is not an important contributor to the $O_3$ distribution, as can be seen in Fig. 14. At this pressure level, stratospheric $O_3$ rich air-masses are kept outside of the AMA circulation. Nevertheless,

on average, STE contributes from 7 to 12 ppbv $O_3$ within the AMA (Table 3). These high values are caused by STE impacting the upper troposphere between 150 hPa and the tropopause, as highlighted by the stratospheric $O_3$ tracer cross-sections in Fig. 13. It is also interesting to note that STE also impacts the free troposphere over the Middle-East and India in a very similar way to African LiNOx, traveling with the westerly winds below the AMA. The same eastward transport of Middle-

East NOx emissions has been shown to slightly (~10%) impact NOx and $O_3$ distributions in the lower troposphere over India (Kunhikrishnan et al., 2006). Nevertheless, as discussed in Section 3, $O_3$ from GEOS-Chem is overestimated in the lower and middle stratosphere (24-90 hPa) by a factor of ~1.7. This overestimation most likely implies a similar overestimation in STE evaluated with the $O_3$ stratospheric tracer and STE is probably responsible for a 4 to 7 ppbv $O_3$ enhancement in the

AMA.

Asian LiNOx therefore appear to be the largest NOx source within the AMA with a contribution to the NOx concentration that is twice to three times larger than South-Asian anthropogenic NOx emissions. This result appears contradictory to that of Kunhikrishnan et al. (2004) who estimated that



during the monsoon in the Indian upper troposphere, 60 to 70 % of NOx come from local surface sources and only 20-25% from LiNOx. This apparent contradiction is due to the fact that Kunhikrishnan et al. (2004) defines the upper troposphere as the 500-150 hPa while the AMA spans the 300-100 hPa domain and, according to Fig. 13, LiNOx have their largest impact between 200 and 100 hPa. Furthermore, the global annual LiNOx source used in Kunhikrishnan et al. (2004) is 2.8 Tg(N)/year

which is in the lower part of the 6±3 Tg(N)/year estimation from Schumann and Huntrieser (2007). In our GEOS-Chem simulations, the global annual LiNOx source is set to 6 Tg(N)/year. Concerning the impact of NOx local sources on the upper tropospheric (500-150 hPa) $O_3$, Kunhikrishnan et al. (2004) found a maximum of 15%. Similar results are found by Kunhikrishnan et al. (2006) with a 10 to 20% sensitivity of $O_3$ to Indian NOx emissions in the middle and upper troposphere (700-200

hPa) over India. From Fig. 14 (g) and (l), we can roughly estimate a production of 9 ppbv in the 500-150 hPa range and 60-95°E by Indian NOx sources. For the same region, we also estimate a rough average of 60 ppbv $O_3$ for the July-August period from Fig. 7 (g) and (j). We have therefore an approximate 15% sensitivity of $O_3$ to the Indian NOx source in good agreement with Kunhikrishnan et al. (2004, 2006). According to Kunhikrishnan et al. (2006), NOx emissions from Indonesia have

a non-negligible effect on upper tropospheric NOx (20-30%) and $O_3$ (10-15%) over India during the ASM period. They also state that the impact of Indonesian emissions is more important over the southern part of India through transport by the tropical easterly jet, which was especially strong in the 1997 El-Nino year. This does not contradict the negligible impact of Indonesian emissions on the AMA composition that we have reported, the AMA being an isolated region north of the tropical

easterly jet.

## 6   Summary and conclusions

In the present study, we have analyzed the CO and $O_3$ distributions and budget in the upper level AMA based on observations from the Metop-A/IASI sensor and on simulations from the global

chemistry transport model GEOS-Chem. Model simulations and spaceborne observations have shown a good general agreement for regional features and the seasonal variations of the upper troposphere distributions, with correlation coefficients of 0.70 for CO and 0.94 for $O_3$. The higher correlation for $O_3$ results from its high variability between the oceanic tropical upper troposphere and the extratropical lower stratosphere. Low CO bias in the lower-middle troposphere has been diagnosed in the

simulations with both spaceborne IASI and MOZAIC in-situ data. Such a bias was already identified by other studies with GC (Liu et al., 2010, 2013). The convective uplift of CO is clearly detected by IASI in the monsoon region but the enhanced upper tropospheric CO resulting from westward transport in the AMA circulation is smoothed over the middle and upper troposphere. For $O_3$, large biases resulting from an accentuation of the S-shape profiles by the AvK smoothing are found over





the tropical oceanic regions.

Based on our IASI observations and model simulations, we have analyzed the CO and $O_3$ distributions in relation with the AMA and monsoon convection. We first developed a method to characterize the 3D boundaries of the AMA based on geopotential height (GH). We found that the AMA could

be defined as the region with GH differences larger than 270 m relative to the GH averaged over the 50°S to 50°N band. Both observations and simulations have revealed an anti-correlation of $O_3$ and CO in the middle and upper troposphere, with lower (resp. higher) $O_3$ (resp. CO) in the eastern part of the domain corresponding to the ASM region than in the western part over the Middle-East, North Africa and the Eastern Mediterranean. This anti-correlation partly results from the convective

uplift of freshly polluted air masses rich in CO but poor in $O_3$ and of the subsidence of $O_3$-enriched and CO poor air masses in the subsidence region in the western part of the domain.

In order to quantify the impact of the different emission sources on the Asian upper troposphere CO and $O_3$ budget, we performed sensitivity simulations with CO and NOx sources switched off by

type and region and one simulation with tagged stratospheric $O_3$. For CO, it appears that South-Asia is the most important contributor (∼25 ppbv) to filling up the AMA because emissions (the IGP), convection and upper-level anticyclone coincide. East-Asia is more polluted than South-Asia but convection in this region is less strong than in South-Asia and does not uplift pollution deep enough into the upper-troposphere to contribute significantly to the AMA CO filling (∼10 ppbv). For the

same reason, NOx from South Asian pollution sources contribute more to the $O_3$ formation within the anticyclone (∼8 ppbv) than NOx from China (∼5 ppbv). Nevertheless, LiNOx from Asia are the most important contributor to the photochemical $O_3$ formation within the AMA with a production which is up to two times larger (10- 14 ppbv) than South Asian pollution. Finally, STE plays an important role for $O_3$ in the upper part of the AMA (above 150 hPa) with a contribution (7-10 ppbv)

which is probably overestimated because of the stratospheric $O_3$ overestimation by the model.

*Acknowledgements.* IASI L1c and L2–EUMETSAT data have been downloaded from the Ether French atmospheric database (http://ether.ipsl.jussieu.fr). The research with IASI is conducted with financial support from the CNES (TOSCA–IASI project). MOZAIC is presently funded by INSU–CNRS, Météo–France, and FZJ

(Forschungszentrum Julich, Germany). The MOZAIC-IAGOS data are available via http://www.iagos.fr/web/, thanks to the support from Ether. MERRA data used in this study were provided by the Global Modeling and Assimilation Office (GMAO) at NASA Goddard Space Flight Center through the NASA GES DISC online archive.



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




**Table 1.** Statistics of GC versus IASI UTLS CO and $O_3$ columns comparison over the $10°$S-$40°$N and 0-$160°$E domain for monthly averages during the MJJASO period. Figures are given for GC profiles smoothed with the averaging kernels (GCwAvK) and figures in italic between brackets correspond to GC raw data.

|  | r | Bias % | Std. Dev. % |
|---|---|---|---|
| CO | 0.70 (0.59) | -11.2 (-11.4) | 9.4 (11.8) |
| $O_3$ | 0.94 (0.93) | -13.8 (-19.6) | 26.5 (32.8) |

**Table 2.** Monthly CO from different sources inside and outside of the AMA in ppbv.

|  | Anthropic East Asia | | Anthropic South Asia | |
|---|---|---|---|---|
|  | AMA | Out | AMA | Out |
| June | 8.3 | 5.3 | 17.3 | 10.1 |
| July | 10.7 | 8.9 | 25.3 | 13.8 |
| August | 9.9 | 10.0 | 23.7 | 15.9 |
| September | 7.8 | 7.4 | 14.3 | 8.2 |

Yan, R. and Bian, J.: Tracing the boundary layer sources of carbon monoxide in the Asian summer monsoon anticyclone using WRF-Chem, Advances in Atmospheric Sciences, 32, 943–951, doi:10.1007/s00376-014-4130-3, 2015.

Zhang, Q., Wu, G., and Qian, Y.: Asia high and its relationship to the climate anomaly over East Asia in summer, J. Meteorol. Soc. Jpn., 80, 733–744, 2002.



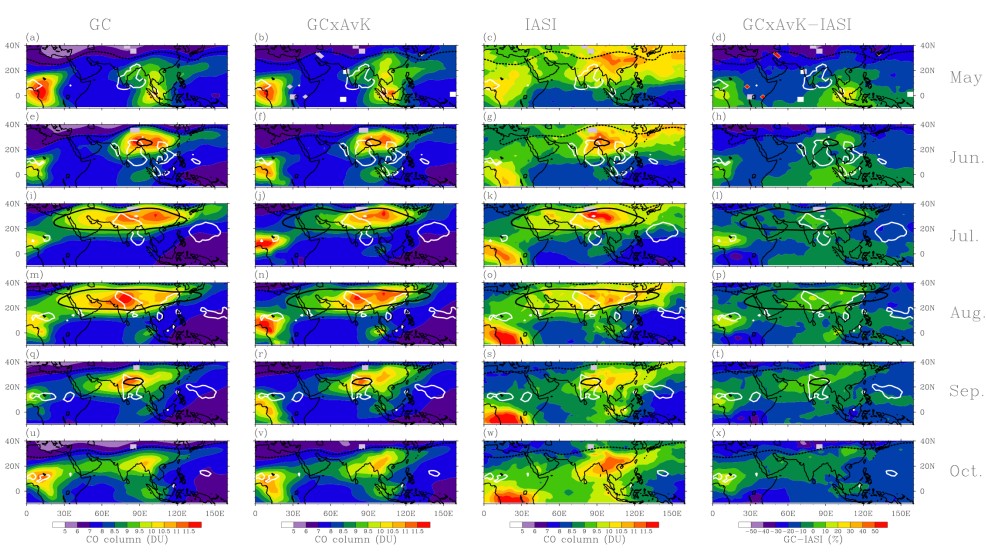

**Figure 1.** Distributions of UTLS (270-110 hPa) CO columns for the May to October 2009 period: (a,e,i,m,q,u) GEOS-Chem, (b,f,j,n,r,v) GEOS-Chem smoothed with IASI AvK, (c,g,k,o,s,w) IASI and (d,h,l,p,t,x) relative differences between GC smoothed with IASI AvK and IASI. From top to bottom, panels correspond to monthly periods with (a,b,c,d) May, (e,f,g,h) June, (i,j,k,l) July, (m,n,o,p) August, (q,r,s,t) September and (u,v,w,x) October. The white solid line represents the 2.5 kg/m$^2$/s Convective Upward Mass Flux from GEOS-5 averaged over 350-150 hPa. The black dashed line is the tropopause (2PVU) and the black solid line is the 12520 m GH representing the AMA boundary at 200 hPa.



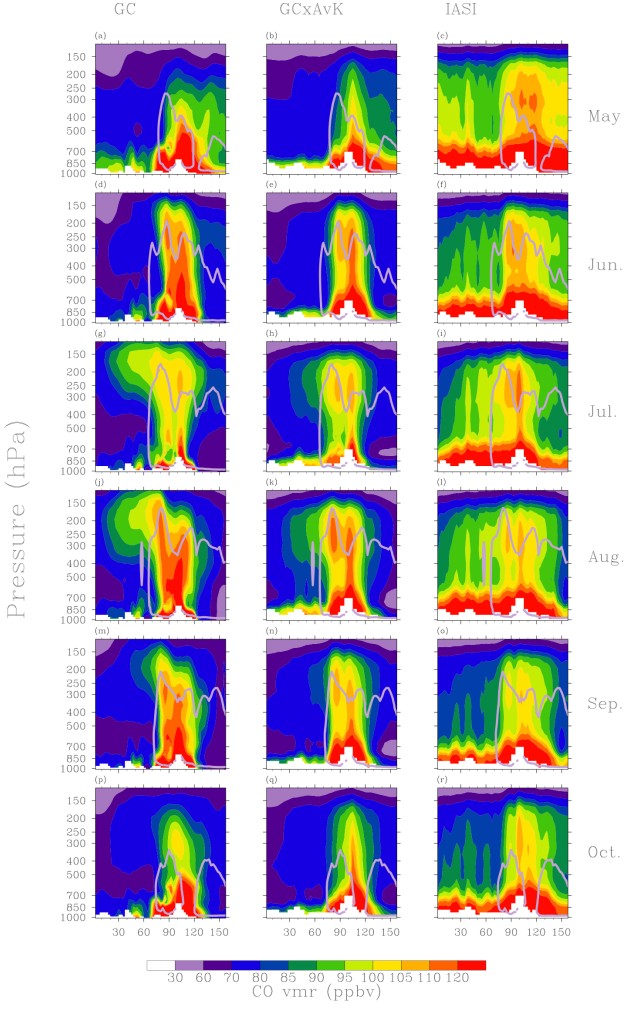

**Figure 2.** Longitude-Pressure cross-sections of CO mixing ratios averaged over 23-29°N for the May to October 2009 period: (a,d,g,j,m,p) GEOS-Chem, (b,e,h,k,n,q) GEOS-Chem smoothed with IASI averaging kernels (c,f,i,l,o,r) IASI. From top to bottom, panels correspond to monthly periods with (a,b,c) May, (d,e,f) June, (g,h,i) July, (j,k,l) August, (m,n,o) September and (p,q,r) October. The grey solid line represents the 2.5 kg/m$^2$/s Convective Upward Mass Flux from GEOS-5.





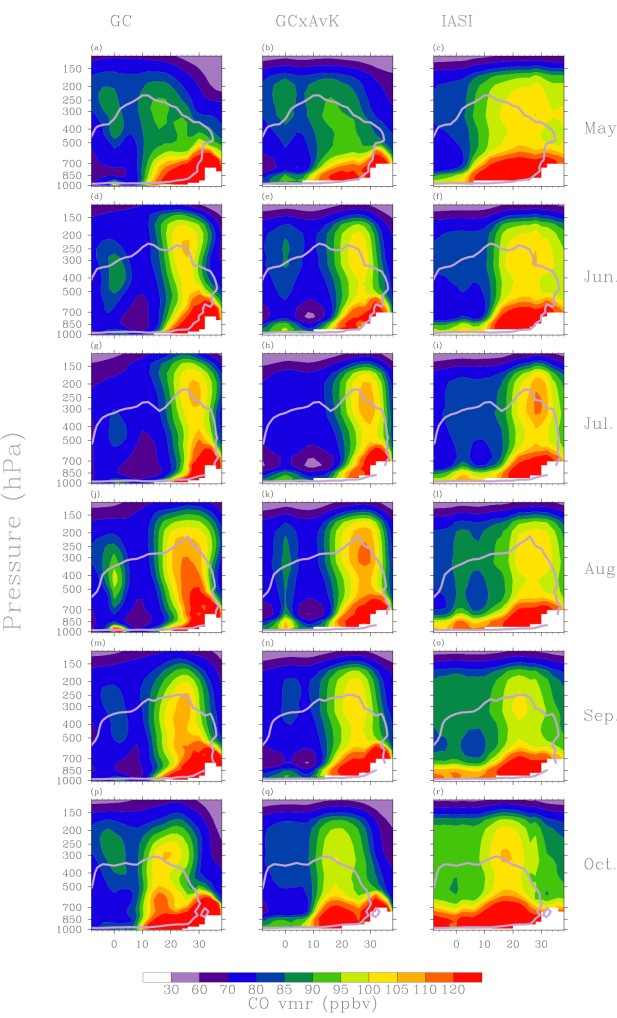

**Figure 3.** Latitude-Pressure cross-sections of CO mixing ratios averaged over 75-105°E for the May to October 2009 period: (a,d,g,j,m,p) GEOS-Chem, (b,e,h,k,n,q) GEOS-Chem smoothed with IASI averaging kernels (c,f,i,l,o,r) IASI. From top to bottom, panels correspond to monthly periods with (a,b,c) May, (d,e,f) June, (g,h,i) July, (j,k,l) August, (m,n,o) September and (p,q,r) October. The grey solid line represents the 2.5 kg/m$^2$/s Convective Upward Mass Flux from GEOS-5.





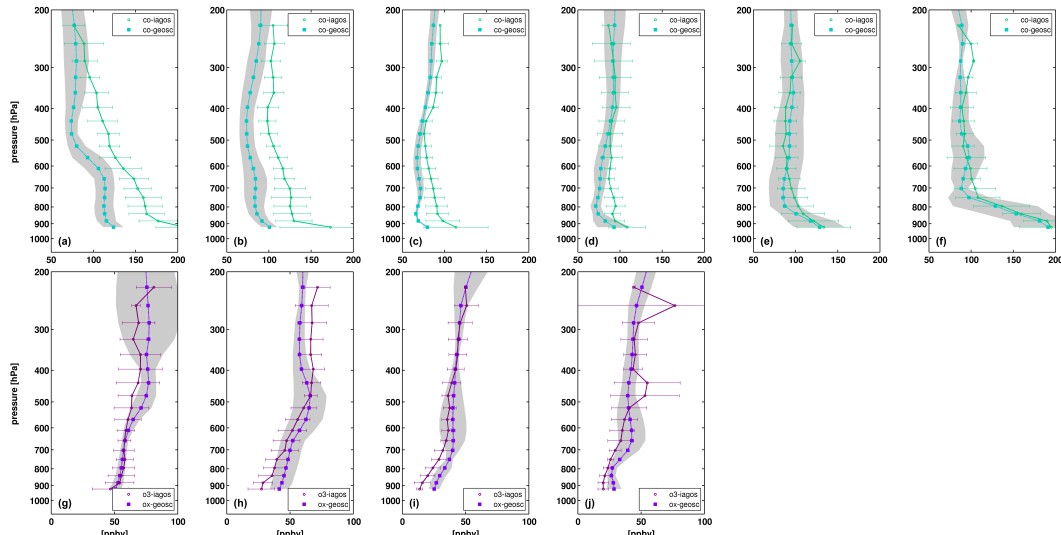

**Figure 4.** Monthly mean tropospheric vertical profiles of CO and $O_3$ at Hyderabad (17.2°N, 78.3°E) from MOZAIC-IAGOS airborne observations and GEOS-Chem simulations. Top panels: CO from May to October 2009, Bottom panels: $O_3$ from May to August 2009. The grey shadings and the error bars represent the 1-$\sigma$ variability for GC and IAGOS respectively.

**Table 3.** Monthly $O_3$ and NOx from different sources inside and outside of the AMA in ppbv. The values for NOx are given in brackets.

|  | Anthropic East Asia | | Anthropic South Asia | | LiNOx Asia | | LiNOx Africa | | Strato. | |
|---|---|---|---|---|---|---|---|---|---|---|
|  | AMA | Out | AMA | Out | AMA | Out | AMA | Out | AMA | Out |
| June | 3.3 | 2.0 | 4.6 | 2.0 | 9.5 | 6.3 | 3.6 | 5.4 | 11.8 | 9.6 |
|  | (0.027) | (0.010) | (0.025) | (0.012) | (0.11) | (0.05) | (0.019) | (0.005) |  |  |
| July | 5.2 | 4.6 | 7.6 | 4.7 | 13.5 | 7.2 | 1.0 | 1.5 | 10.0 | 5.9 |
|  | (0.033) | (0.021) | (0.043) | (0.017) | (0.129) | (0.048) | (0.012) | (0.018) |  |  |
| August | 4.9 | 5.2 | 8.1 | 4.4 | 9.9 | 5.3 | 0.9 | 1.2 | 6.7 | 3.9 |
|  | (0.027) | (0.023) | (0.042) | (0.016) | (0.087) | (0.036) | (0.011) | (0.022) |  |  |
| September | 3.4 | 3.3 | 5.2 | 4.0 | 6.1 | 5.0 | 1.1 | 2.2 | 6.7 | 4.7 |
|  | (0.018) | (0.014) | (0.033) | (0.018) | (0.074) | (0.044) | (0.010) | (0.032) |  |  |





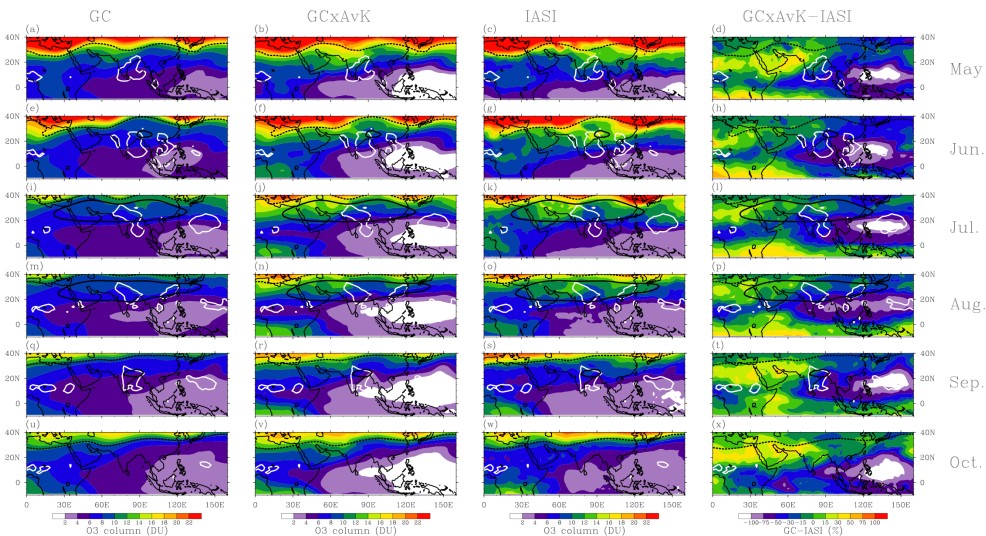

**Figure 5.** Same as Figure 1 for $O_3$.

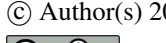



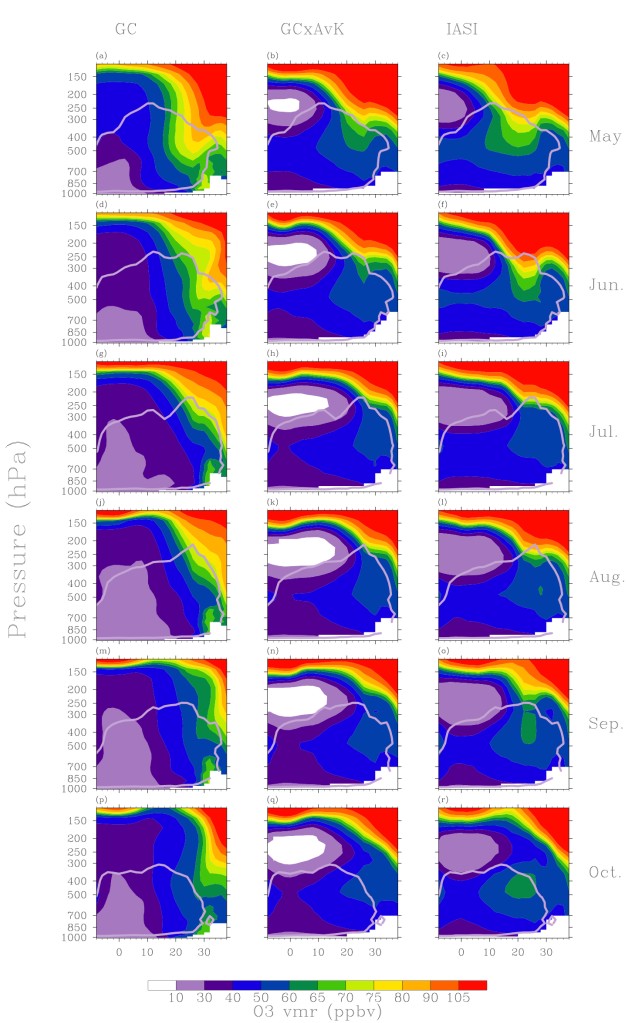

**Figure 6.** Same as Figure 3 for $O_3$.





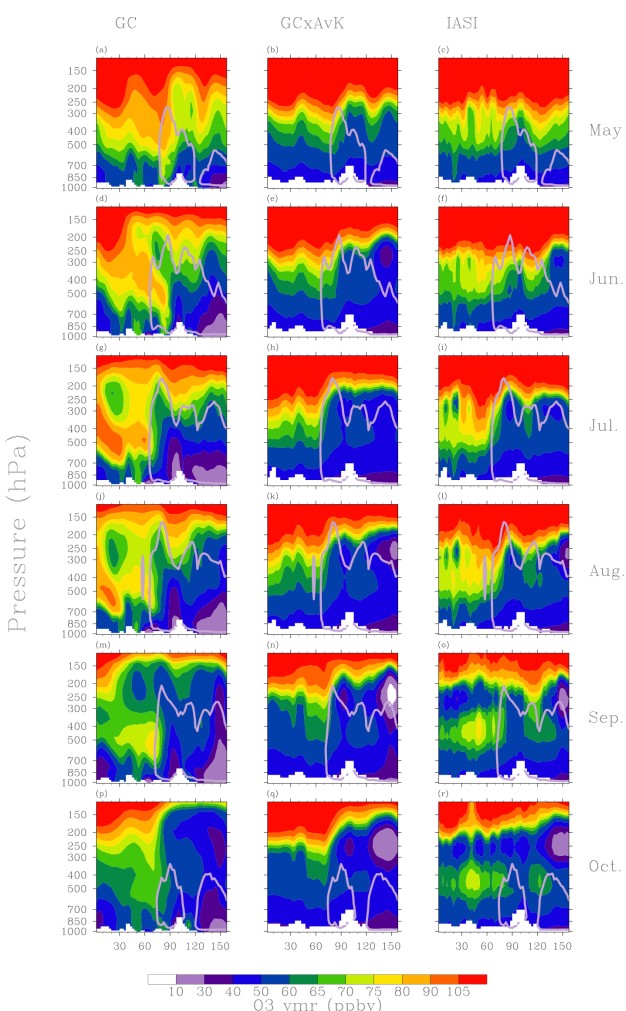

**Figure 7.** Same as Figure 2 for $O_3$.





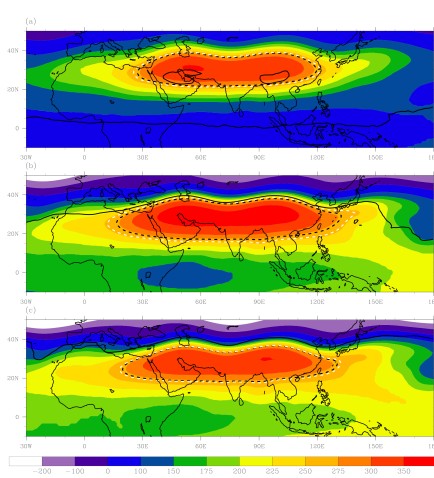

**Figure 8.** Geopotential heights (GH) from MERRA for July 2009 at (a) 100 (b) 150 hPa and (c) 200 hPa. The black dotted lines represent the GH isocontours at (a) 16770 m (b) 14350 m and (c) 12520 m and the white dotted line represents the 270 m GH anomalies (see text for details).





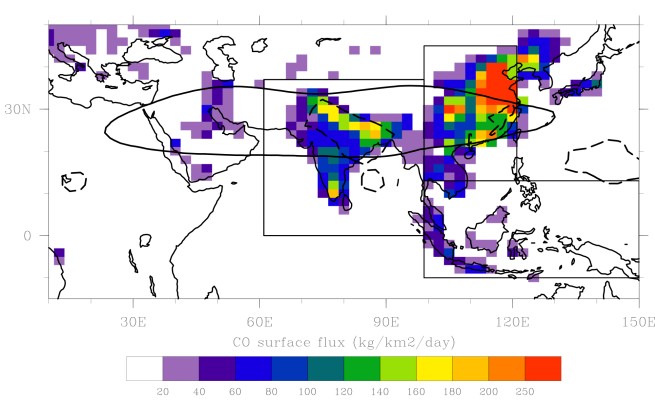

**Figure 9.** Anthropogenic emissions of CO from the Streets 2006 inventory for July. The black dashed line is the 2.5 kg/m$^2$/s Convective Upward Mass Flux contour at 2225 hPa from GEOS-5 for July 2009 and the solid black line is the 12520 m GH contour from MERRA at 200 hPa for July 2009. The 3 boxes correspond to the regions selected for the sensitivity simulations with anthropogenic emissions switched off (South, East and South-East Asia).





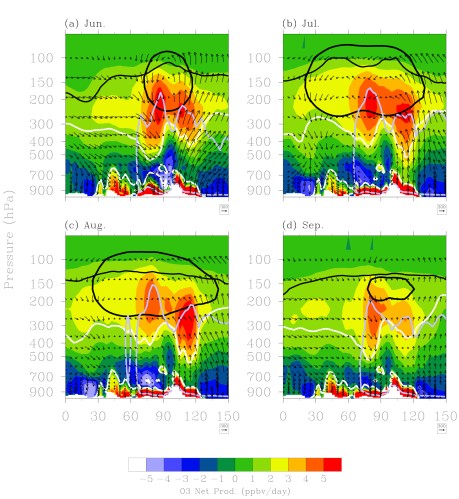

**Figure 10.** Longitude-pressure cross-sections of GC simulated $O_3$ net production rates averaged over 23-29°N in (a) June, (b) July, (c) August and (d) September 2009. The black arrows correspond to the $O_3$ fluxes and the white solid lines to the 100 pptv NOx contours from GC. The dashed black line corresponds to the tropopause (2 PVU), the grey solid line to upward convective mass fluxes of 2.5 kg/m$^2$/s at 200 hPa and the black solid line to the AMA boundary computed as the 270 m GH anomaly (see text for details).





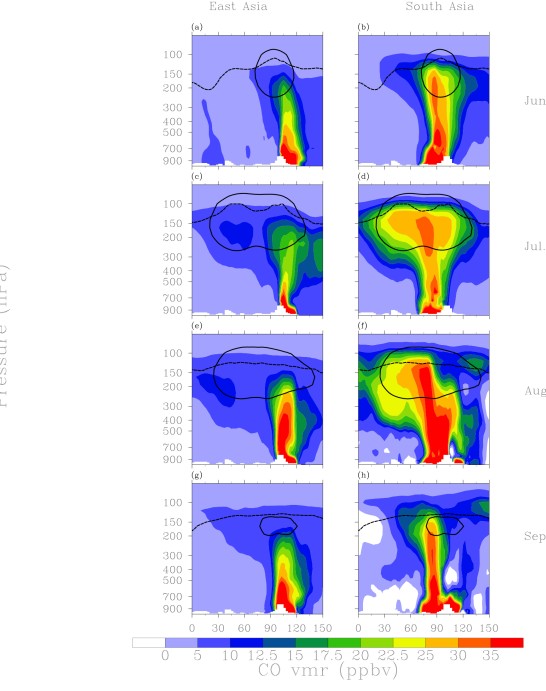

**Figure 11.** Longitude pressure cross sections of the sensitivity of CO to anthropogenic CO sources averaged over 23-29°N from (a,c,e,g) East Asia (b, d,f,h) South Asia computed as the differences between the control run and simulations with the corresponding source switched off. From top to bottom, panels correspond to (a,b) June, (c,d) July, (e,f) August and (g,h) September 2009.





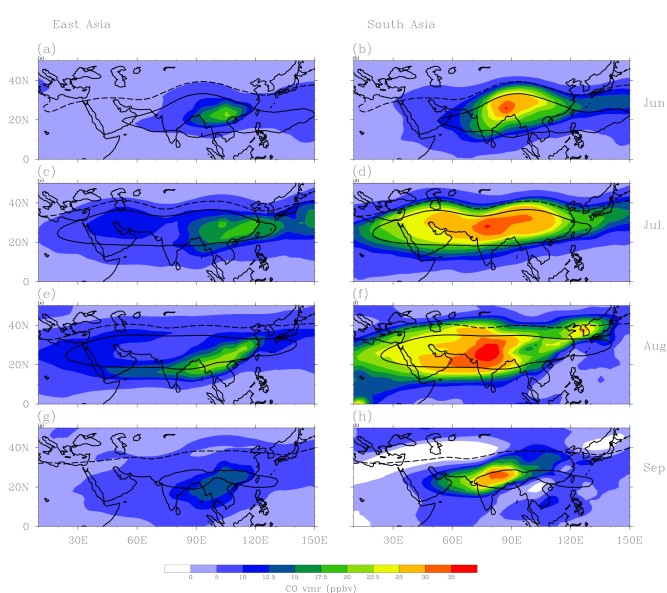

**Figure 12.** Same as in Fig. 11 for the distributions at 200 hPa.





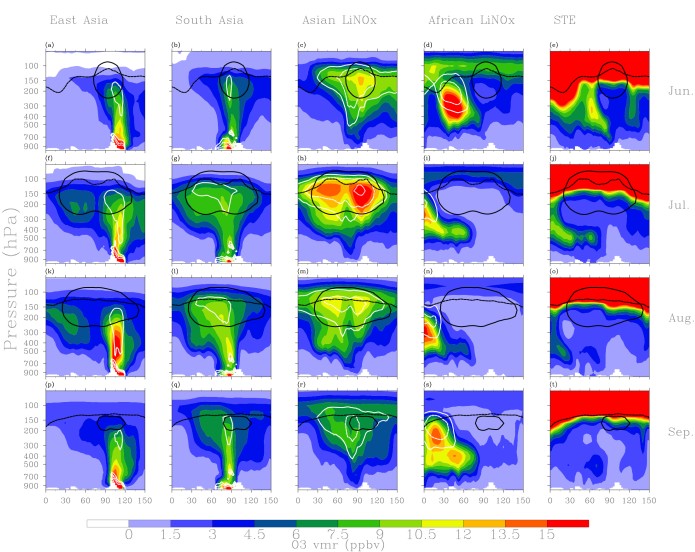

**Figure 13.** Longitude pressure cross sections of the sensitivity of $O_3$ to NOx sources averaged over 23-29°N for: (a,f,k,p) East Asian anthropogenic (b,g,l,q) South Asian anthropogenic (c,h,m,r) Asian lightning (d,i,n,s) African lightning computed as the difference between the control run and simulations with the corresponding source switched off. Panels (e,j,o,t) correspond to tagged stratospheric $O_3$ to diagnose STE. From top to bottom, panels correspond to monthly means with (a,b,c,d,e) June, (f,g,h,i,j) July, (k,l,m,n,o) August and (p,q,r,s,t) September. The white solid, dashed and dotted lines correspond to the 50, 100 and 200 pptv contours for the sensitivity of NOx to the different NOx sources.




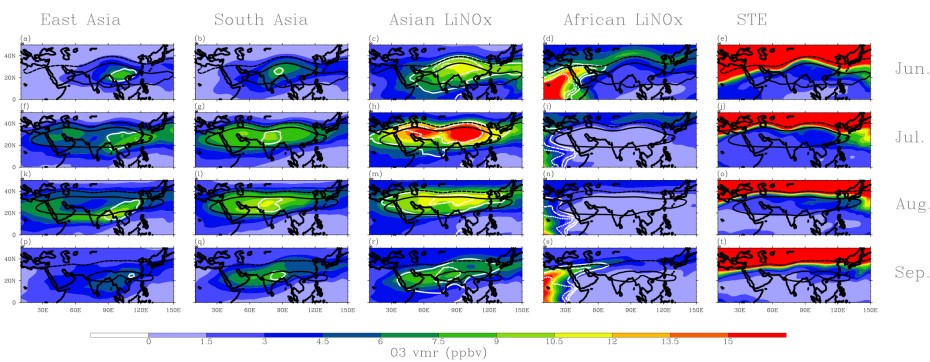

**Figure 14.** Same as in Fig. 13 for the distributions at 200 hPa.