# Peer review of "Upper tropospheric CO and O3 budget during the Asian Summer Monsoon"

_Atmospheric Chemistry and Physics, 2015_

## Referee Comment (RC1) · Anonymous Referee #1 · 20 Feb 2016

This paper presents an evaluation of the CO and O3 budgets in the Asian Summer Monsoon region, based on simulations using the GEOS-Chem model. About half of the paper is devoted to evaluating the model chemical climatology using IASI satellite and MOZAIC/IAGOS aircraft measurements, and the comparisons show reasonable seasonal and spatial chemical behavior for the model. It is good to see IASI constituent data being utilized, and the comparisons with the model are made using the appropriate averaging kernels. The model is then used to perform sensitivity tests to quantify source regions and chemical budgets for the monsoon UTLS region, with results focusing on CO, NOx and ozone production rates. The overall chemical budget calculations and results seem reasonable. Chemical behavior in the monsoon region is a topic of substantial current interest, and the results here contribute to understanding the details of the GEOS-Chem simulation and chemistry and transport in the real atmosphere. Overall the paper is reasonably well written and the authors have done a good job of including numerous references to previous work. I recommend this paper for publication in ACP, but have several comments for the authors to consider in revision.

1) One overarching comment is that I personally disliked papers that include numerous figures with many small panels ('postage stamps'), for which the reader is expected to scrutinize details in each of the panels. Figures 1,2,3,5,6,7,13 and 14 are such figures in this paper, showing detailed evolution of various diagnostics during May-October. I would recommend an alternative methodology of showing one or two key months in each of these figures, with enlarged scale to allow focus on the important details. The seasonal evolution can be described in words, and the entire sequence could be included in Supplementary material if necessary.

2) One detail that I don't understand regards the appearance of the 'S-shaped' ozone profile in the GCxAvK calculations, which don't appear in the GC model itself (Fig. 6). I don't understand this because the averaging kernels are broad in the vertical (6-8 km), and so how can they introduce narrow vertical structure into the weighted model results? Is this possibly due to the a priori profiles that are also used in the calculations?

3) Correlation coefficients are often quoted in comparing IASI vs. model results. Do these refer to spatial or temporal correlations?

4) There are numerous English errors in the text that should be corrected. Also, Fig. 10 is called out before Fig. 9.

---

## Referee Comment (RC2) · Anonymous Referee #2 · 25 Feb 2016

**General comment:** This paper addresses the chemical budgets of ozone and CO during the Asian monsoon, based on a comparison between observations from IASI, IAGOS and model simulations. Particular focus is laid on the budget within the upper tropospheric monsoon anticyclone and effects on the buildup of the tropospheric ozone enhancement over the Middle East. Sensitivity simulations are used to isolate the effects of different source regions. For CO as well as for ozone (via NOx-precursors) South Asia is found to have a stronger contribution on the composition of the anticyclone, compared to emissions from East Asia, but lightning-NOx is identified as the largest contributor to ozone formation. The tropospheric ozone maximum over the Middle East is related to downwelling ozone fluxes to the west of the monsoon.

Overall, the paper is well written and addresses a topic of interest for the ACP-readership. In the following, I have two major and several minor comments which

should be considered before publication.

**Major comments:**

1) Asian monsoon boundary:

My first main concern is related to the separation between the monsoon anticyclone interior and its surroundings, based on geopotential height, as used in this paper. I personally think that PV would be a more suitable quantity describing the confinement of air masses in the anticyclone. At least, there are some recent papers showing that trace gas contours in the anticyclone align more closely with PV than geopotential, and that enhanced PV-gradients even indicate the existence of a transport barrier (e.g., Garny and Randel, 2013; Ploeger et al., 2015; Garny and Randel, 2015).

Presumably, the results presented here are not very sensitive to the usage of either GH or PV, as values are always calculated for the whole anticyclone (e.g., Table 3) and differences in the total area (defined by either PV or GH) are not very large. However, it would be nice to have some sensitivity study quantifying the uncertainty due to using either GH or PV. At least a thorough discussion about defining the anticyclone boundary and a reasoning why a geopotential anomaly is used here, should be included. (There are already some related text parts in Sect. 4.1, but these could be extended).

A related question is: Have daily GH fields been used for defining the anticyclone and calculating the fractions (e.g., Table 3), or the monthly mean as plotted in the figures? I would strongly suggest to do the latter, if this has not already been done.

2) Middle East ozone maximum:

I'm confused about the discussion concerning the buildup of the ozone maximum over the Middle East. It is concluded (e.g., Sect. 4.2) that this ozone maximum is largely related to downward flux from the Asian monsoon. However, Fig. 7 shows that outflow from the monsoon in the upper troposphere to the west during July–September lowers ozone mixing ratios at levels between 400–150hPa over the Middle East (e.g., Fig. 7j).

Below, at levels of 700–400 hPa, ozone mixing ratios remain high. In my opinion, these higher values are related to African LiNOx and STE and not to transport from within the anticyclone, as Fig. 13 shows. I think the reasoning here needs to be clarified.

**Minor comments:**

P2, L36: I wouldn't consider the Asian monsoon as an extra-tropical phenomenon, but rather subtropical or even tropical (e.g., Fueglistaler et al., 2009).

P3, L1: How is this fact (*...convection from the Tibetan Plateau, highlighted as predominant to fill the AMA*) related to the recent analysis by Tissier et al. (2015), stating that "the Tibetan plateau ... (is) a minor overall contributor..."?

P3, L78: *...high altitude...* Which altitude?

P14, 476: *These combined effects...* I think the dominant effect causing the low ozone anomaly in the Asian monsoon is vertical transport. Models without tropospheric ozone chemistry included do a reasonably good job in simulating the low ozone concentrations in the monsoon anticyclone (e.g., Konopka et al., 2010).

P15, 495: Figure 10 shows upward mass fluxes at both peaks in ozone pruduction rates (also for the western one). Therefore, I would say *...the double maximum...is associated with a double peak structure in upward mass flux.*

Fig. 14 / and related discussion: The fact that South Asian emissions fill out the anticyclone whereas East Asian emissions are transported around seems very consistent with the findings of Vogel et al. (2015) regarding the transport from various surface regions to the anticyclone.

**Technical corrections:**

Title: O3 → $O_3$

P4, L113: (Clerbaux et al. (2009)) → (Clerbaux et al., 2009)
Similar wrong bracketing occurs several times throughout the paper (e.g., P4/L126, ...).

P6, L182:  no number before °S.

P7, L219:  10S° → 10°S

P8, L259:  Here, the latitude band is defined to be 21–29°N, while in the figure caption it is written to be 23–29°N. This discrepancy occurs also later in the paper several times.

P8, L260:  correspondS

P10, L312:  Should be *equation (1)*.

P11, L362:  highlights → highlight

P11, L378:  Shouldn't this read *...eastern part...*?

P14, L453:  thann → than

P14, L470:  ...convective uplift of...

P14, L471:  resultS

Fig. 9 / caption:  I guess the level for the mass flux contour should read 225 hPa.

**References:**

Fueglistaler et al. (2009), Rev. Geophys., 47, RG1004.

Garny and Randel (2013), J. Geophys. Res, 118, 13421–13433.

Garny and Randel (2015), Atmos. Chem. Phys. Discuss., 15, 25981–26023.

Konopka et al. (2010), Atmos. Chem. Phys., 10, 121–132.

Ploeger et al. (2015), Atmos. Chem. Phys., 15, 13145–13159.

Vogel et al. (2015), Atmos. Chem. Phys., 15, 13699–13716.

Tissier et al. (2015), Atmos. Chem. Phys. Discuss., 15, 26231–26271.

---

## Author Comment (AC1) · 6 Jun 2016

**Reply to reviewer 1:**
We first thank Reviewer 1 for his suggestions to improve our paper.

**Comment 1:**
*One overarching comment is that I personally disliked papers that include numerous figures with many small panels ('postage stamps'), for which the reader is expected to scrutinize details in each of the panels. Figures 1,2,3,5,6,7,13 and 14 are such figures in this paper, showing detailed evolution of various diagnostics during May-October. I would recommend an alternative methodology of showing one or two key months in each of these figures, with enlarged scale to allow focus on the important details.*

[Figure]

*The seasonal evolution can be described in words, and the entire sequence could be included in Supplementary material if necessary.*

We agree with the reviewer: the panel plots contain too many panels which make them unecessarily heavy. Nevertheless, in order to make the monsoon impact on the composition clear, we need to show more than one or two months. For the general context and the model validation with IASI (Fig. 1, 2, 3, 5, 6, 7), we have chosen to keep 3 months out of 6: May (pre-monsoon), July (monsoon) and October (post-monsoon). The plots show very different structures that are important to see on 2D plots. For the budget plots (Fig. 11, 12, 13, 14), we have kept June, July and September to show the evolution during the monsoon itself. As August is very similar to July we have eliminated the August plots. The results from the whole monsoon period are still summarized in Table 2.

**Comment 2:**
*One detail that I don't understand regards the appearance of the 'S-shaped' ozone profile in the GCxAvK calculations, which don't appear in the GC model itself (Fig. 6). I don't understand this because the averaging kernels are broad in the vertical (6-8 km), and so how can they introduce narrow vertical structure into the weighted model results? Is this possibly due to the a priori profiles that are also used in the calculations?*

The O3 profiles are naturally S-Shaped in the tropics through the effect of convection which reduces the UT concentrations. The convolution with the AvK accentuates the S-Shape by reducing even more the concentrations in the tropical UTLS as seen in Fig. 6. As mentioned in the text this effect has ben discussed in Dufour et al. 2012 with comparisons between IASI and ozonesondes. The effect is not narrow as the reviewer mentions but spans the whole tropical UT from 400 to 200 hPa (8 to 12 km,

Fig. 6, panels b, e, h) in agreement with the width of the AvK (6-8 km).

**Comment 3:**
*Correlation coefficients are often quoted in comparing IASI vs. model results. Do these refer to spatial or temporal correlations?*

They refer to spatio-temporal correlations for the data plotted in Fig. 1 (CO) and 5 (O3) concerning UTLS columns monthly averaged and gridded (for IASI) on the GEOS-Chem grid. Now that we have elliminated 3 out of the 6 months shown in these figures we have modified the text accordingly.
**"The statistics of the CO UTLS columns comparison (for the domain displayed in Fig. 1 and the 6 months from May to October) are summarized in Table 1"**

**Comment 4:**
*There are numerous English errors in the text that should be corrected. Also, Fig. 10 is called out before Fig. 9.*

The text has been proof-read by a native speaker profesionnal in proof-reading and translations of scientific publications. We hope that most of the errors are gone! Figures 10 and 9 have been reordered.

---

## Author Comment (AC2) · 6 Jun 2016

**Reply to reviewer 2:**

We thank reviewer 2 for his/her carefull reading and for his/her comments and suggestions that helped us improve our manuscript.

**Major comments:**
**Comment 1:**
*My first main concern is related to the separation between the monsoon anticyclone interior and its surroundings, based on geopotential height, as used in this paper. I personally think that PV would be a more suitable quantity describing the confinement*

[Figure]

*of air masses in the anticyclone. At least, there are some recent papers showing that trace gas contours in the anticyclone align more closely with PV than geopotential, and that enhanced PV-gradients even indicate the existence of a transport barrier (e.g., Garny and Randel, 2013; Ploeger et al., 2015; Garny and Randel, 2015). Presumably, the results presented here are not very sensitive to the usage of either GH or PV, as values are always calculated for the whole anticyclone (e.g., Table 3) and differences in the total area (defined by either PV or GH) are not very large. However, it would be nice to have some sensitivity study quantifying the uncertainty due to using either GH or PV. At least a thorough discussion about defining the anticyclone bound- ary and a reasoning why a geopotential anomaly is used here, should be included. (There are already some related text parts in Sect. 4.1, but these could be extended).*

The studies mentioned by the reviewer and older ones (Barret et al. 2008) indeed show that PV is highly correlated with tracer concentrations and allow a good determination of the AMA boundaries on a daily time scale. Nevertheless, we are analysing monthly averages and budgets which are not much dependent on the fine structure better detected by PV than by GH. Furthermore, Ploeger et al. (2015) is the only study that proposes a PV-based criterion to define the AMA boundaries on a daily scale but the criterion is only validated for the 380 K level (200 hPa). Many studies have defined robust GH thresholds to delimit the AMA which agree very well with the gaz tracer concentration contours on monthly timescales. Because we need a 3D criterion valid for the whole UTLS altitude range on a monthly timescale, we have prefered to build a GH-based criterion. As suggested by the reviewer we have therefore included "a thorough discussion about defining the anticyclone bound-ary and a reasoning why a geopotential anomaly is used here" in section 4.1:

**"Based on MLS CO analyses Barret et al. (2008) have shown that daily CO and PV variations were strongly correlated with low PV related to high CO. In the**

ama, the tracer concentration is therefore strongly controlled by the oscillations and sheddings of the AMA. In their study about the AMA strength and variability, Garny and Randel (2013) have also pointed to the spatio-temporal correlation of CO enhancements and low PV values which is stronger in the upper levels of the AMA. Based on PV fields Ploeger et al. (2015) have developed a method to characterize the dynamical barrier that delimit the inside and the outside of the AMA on a daily timescale. The boundaries of the AMA based on their method are consistent with tracer concentrations (high CO and low O3 within the AMA). In studies looking at monthly or seasonal timescales, the edge of the AMA has been mostly defined as simple constant GH contours at different pressure levels. Randel et al. (2006) (resp. Heathand Fuelberg (2014)) use a 14320 (resp. 14430) m GH for the AMA at 150 hPa and Bergman (2013) use 12520 (resp. 16770) m GH at 200 (resp. 100) hPa. In order to determine the CO and O3 budget within the AMA, we first need to characterize the AMA as a closed volume and we have therefore looked for a criterion independent of the pressure level. As already discussed, the studies based on PV (Barret et al., (2008), Garny et al. (2013) , Ploeger et al. (2015)) have shown that it was a good dynamical parameter to charactrize the AMA high frequency variability whilst GH was mostly used on monthly timescales (Randel et al. (2006) , Bergman et al. (2013), Heath and fuelberg (2014). Furthermore, Ploeger et al. (2015) is the only study that proposes a PV-based criterion to delimit the AMA but this criterion is only defined and validated for the 380 K potential temperature level ( 200hPa). As the PV tracer relationship is stronger at the higher levels (380K) of the AMA (Garny and Randel (2013)) the criterion from Ploeger et al. (2015) may not hold for the lower levels. Finally, on monthly timescales, simple GH thresholds have been shown to consistently delimit regions of tracer anomalies characteristic of the AMA at different pressure levels. We have therefore chosen to use a criterion based on GH rather than PV to delimit the AMA. Our criterion is based on thresholds of GH anomalies."

*A related question is: Have daily GH fields been used for defining the anticyclone and calculating the fractions (e.g., Table 3), or the monthly mean as plotted in the figures? I would strongly suggest to do the latter, if this has not already been done.*

We have used the MERRA monthly GH fields. This is now mentionned in the text.

**Comment 2:**
*I'm confused about the discussion concerning the buildup of the ozone maximum over the Middle East. It is concluded (e.g., Sect. 4.2) that this ozone maximum is largely related to downward flux from the Asian monsoon. However, Fig. 7 shows that outflow from the monsoon in the upper troposphere to the west during July–September lowers ozone mixing ratios at levels between 400–150hPa over the Middle East (e.g., Fig. 7j).Below, at levels of 700–400 hPa, ozone mixing ratios remain high. In my opinion, these higher values are related to African LiNOx and STE and not to transport from within the anticyclone, as Fig. 13 shows. I think the reasoning here needs to be clarified*

We agree with the reviewer that the statements concenrning the origin of the O3 high concentrations over the Middle East have to be clarified even if it is not the focus of our paper. The important point is that the origin of O3 depends on the altitude range. As mentioned by the reviewer and shown in fig. 7d (O3 in July) the O3 below 400 hPa seems to be coming from the west. This is corroborated by Fig. 13 which shows that African LiNOx (panel i) and STE (panel j) are largely contributing to the O3 enhancement below the AMA, between 600 and 400 hPa. As mentioned by the reviewer the AMA circulation is lowering the O3 concentration in the UT (280-150 hPa) over the Middle-East in July relative to the other months as shown by Figure 7 because

it is recirculating O3 poor convective air masses. Nevertheless, according to Figure 13, southern Asian anthropogenic emissions (13 g) and Asian LiNOX (13h) are enhancing O3 between 400 hPa and the lower limit of the AMA through the downward O3 fluxes displayed in Fig. 9b (July). We have therefore modified the manuscript as follows

In section 4.2, the O3 Middle-East is only mentioned but we have changed " largely" to "partly" in the statement and we refer to section 5.2 (O3 budget) where it is discussed in more details:

"As already discussed, this downward flux **partly** contributes to the build-up of the Middle East O3 maximum as described in Liu et al. (2009) **and discussed in section 5.2**."Âă

In section 5.2 the text has been modified as followsÂă:

ÂńÂăDuring the July-August period, the large subsidence over the Middle-East (30-60E) (Fig. 9, (b) and (c)) brings O3 produced by both South Asian anthropogenic NOx and Asian LiNOx **down to 400 hPa (Fig. 13 (g) and (h))** and contributes to **the upper part** of the mid-tropospheric O3 maximum. **Below 400 hPa and down to 600 hPa, air masses coming from the West are not blocked by the AMA and both African LiNOx and STE have a larger contribution to the free tropospheric Middle-East O3 maximum (Fig. 13 (i) and (j)) highlighted by GC and IASI (Fig. 7 (d) and (f)) than Asian sources.**ÂăÂż

**Minor comments:**

**P2, L36:** I wouldn't consider the Asian monsoon as an extra-tropical phenomenon, but rather subtropical or even tropical (e.g., Fueglistaler et al., 2009).

That's right. We have changed to tropical.

**P3, L1:** *How is this fact (...convection from the Tibetan Plateau, highlighted as predominant to fill the AMA) related to the recent analysis by Tissier et al. (2015), stating that "the Tibetan plateau ... (is) a minor overall contributor..."?*

The complete Tissier et al. (2015) statement is ÂńÂăThe Tibetan plateau, although a minor overall contributor, is found to be the region with the highest impact of convection at 380 K due to its central location beneath the Asian upper level anticyclone.ÂăÂż There is no contradiction with our statement ÂńÂăthe Tibetan plateau is predominant to fill the AMA.ÂăÂż based on Bergman et al. (2013) and Heath and Fuelberg (2014). Indeed, Tissier et al. (2015) look at air masses reaching the TTL at the global scale (ÂńÂăoverallÂăÂż) while in the two other studies they look at the AMA.

**P3, L78:** *...high altitude... Which altitude?*

Our statement is "Based on in-situ data recorded at the Himalayan NCO-P observatory, Cristofanelli et al. (2010) have shown that high altitude" which implies that high altitude is the altitude of the observatory. We have added the precise altitude (5049 m).

**P14, 476**: *These combined effects... I think the dominant effect causing the low ozone anomaly in the Asian monsoon is vertical transport. Models without tropospheric ozone chemistry included do a reasonably good job in simulating the low ozone concentrations in the monsoon anticyclone (e.g., Konopka et al., 2010).*

The reviewer misunderstood the statement. In this paragraph, we are dealing with the O3 difference between South Asia and the Middle East in the mid-troposphere and not with the O3 difference between the AMA (which encompasses much larger area than South Asia) and the rest of the tropical UT which is clearly a result of transport to the first order. Therefore transport is not the dominat factor in lowering O3 and clouds and precursor transport are also important. We have therefore added mid-tropospheric in the statementÂǎ:

"These combined effects are responsible for the lower **mid-tropospheric** O3 concentrations over South Asia compared to regions with high insolation and downward transport of O3, such as the Middle East and North Africa.Âǎ"

**P15, 495**: *Figure 10 shows upward mass fluxes at both peaks in ozone pruduction rates (also for the western one). Therefore, I would say ...the double maximum...is associated with a double peak structure in upward mass flux.*

We do not agree with the reviewer, for the months of June, July and September the westernmost O3 production peak is associated with a downward O3 flux and almost no vertical flux in August. We have therefore modified the text as follows:

**"Âǎ...by a double maximum with values exceeding 5 ppbv/day that are associated with the upward fluxes east of 90E and downward fluxes (except in August) west of 90E.Âǎ"**

**Fig. 14** *and related discussion: The fact that South Asian emissions fill out the anticy-*

*clone whereas East Asian emissions are transported around seems very consistent with the findings of Vogel et al. (2015) regarding the transport from various surface regions to the anticyclone.*

Vogel et al. (2015) use lagrangian simulations with a BL tracer. We therefore think it is more suitable to discuss their paper with Fig. 12 displaying CO (mostly a primary pollutant) than with Fig. 14 displaying O3 (secondary pollutant). It is rather difficult to compare our results to those of Vogel et al. (2015) because the selected emission regions are different. Especially their South Asian region encompasses ours and part of our East Asian region. Nevertheless, we think it is an interesting comparison and we have added the following statements in the discussion of the CO budget:

**"Vogel et al. (2015) have also quantified the origin of PBL air masses in the AMA using artificial emission tracers from the CLaMS CTM. Their emission regions are different from those used in the present study. India is separated in Northern and Southern India and South East Asia encompasses our South East Asia and part of our East Asia (most of the Indochina peninsula). Nevertheless, their results show some agreement with ours and give some complementary information. They show that when the AMA is established, PBL airmasses coming from Northern India are filling up the AMA comparably to our South Asian tracer which indicates that South Asia plays a minor role. Their South East Asian emission tracer is transported upwards and remains at the edge of the AMA such as our East Asian tracer (especially for August, not sh own). The agreement probably comes from the fact that both tracers encompasses the Indochinese peninsula where convection is strong during the monsoon but which is located to the south of the AMA.Âă"**